# Unraveling Histone Loss in Aging and Senescence

**DOI:** 10.3390/cells13040320

**Published:** 2024-02-09

**Authors:** Sushil Kumar Dubey, Rashmi Dubey, Mark Ellsworth Kleinman

**Affiliations:** Department of Surgery, East Tennessee State University, Johnson City, TN 37614, USA; dubeys@etsu.edu (S.K.D.); dubeyr@etsu.edu (R.D.)

**Keywords:** histones, nucleosome occupancy, aging, senescence, epigenetics, histone degradation, histone variant

## Abstract

As the global population experiences a notable surge in aging demographics, the need to understand the intricate molecular pathways exacerbated by age-related stresses, including epigenetic dysregulation, becomes a priority. Epigenetic mechanisms play a critical role in driving age-related diseases through altered gene expression, genomic instability, and irregular chromatin remodeling. In this review, we focus on histones, a central component of the epigenome, and consolidate the key findings of histone loss and genome-wide redistribution as fundamental processes contributing to aging and senescence. The review provides insights into novel histone expression profiles, nucleosome occupancy, disruptions in higher-order chromatin architecture, and the emergence of noncanonical histone variants in the aging cellular landscape. Furthermore, we explore the current state of our understanding of the molecular mechanisms of histone deficiency in aging cells. Specific emphasis is placed on highlighting histone degradation pathways in the cell and studies that have explored potential strategies to mitigate histone loss or restore histone levels in aging cells. Finally, in addressing future perspectives, the insights gained from this review hold profound implications for advancing strategies that actively intervene in modulating histone expression profiles in the context of cellular aging and identifying potential therapeutic targets for alleviating a multitude of age-related diseases.

## 1. Introduction

With demographic shifts toward aging populations, the incidence of age-related diseases such as cancer, diabetes, cardiovascular disorders, osteoarthritis, and macular degeneration is on the rise in many countries [1,2,3]. There is a massive unmet need to significantly improve our fundamental scientific understanding of aging biology in order to develop new therapeutic approaches to age-related diseases. Achieving this goal hinges on decoding the intricate molecular mechanisms of aging and identifying new experimental models to mitigate or reverse age-associated phenotypes.

Aging is a complex and multifactorial process characterized by a combination of aging hallmarks that contribute to declines at the molecular, cellular, and systemic levels in an organism. The hallmarks of aging, as outlined by López-Otín et al. [4], are categorized into three groups: primary hallmarks, antagonistic hallmarks, and integrative hallmarks. The primary hallmarks of aging that are deleterious to the cell include genomic instability, telomere attrition, epigenetic alterations, loss of proteostasis, and disabled macroautophagy [5,6,7,8,9,10,11]. The antagonistic hallmarks represent a more intricate response of the cell to aging and include deregulated nutrient sensing, cellular senescence, and mitochondrial dysfunction. The cellular nutrient sensing pathways that adapt to the nutrition availability change with age. Deregulated nutrient sensing is closely associated with mitochondrial dysfunction, peroxisomal dysfunction, and the deregulation of protein synthesis and glucose, nucleotide, and lipid metabolisms [12,13]. Mitochondrial dysfunction can also induce oxidative stress, which can initially benefit the cell by eliciting a protective gene response but may prove detrimental in the long run [14,15]. Collaborating closely with mitochondria, peroxisomes play a crucial role in preserving the oxidative balance within the cell [16]. While the peroxisomes typically generate elevated levels of reactive oxygen species and reactive nitrogen species, they also are armed with a battery of antioxidant enzymes and nonenzymatic free radical scavengers [17]. In addition to regulating ROS metabolism, peroxisomes perform crucial functions in lipid metabolism. However, a growing body of evidence indicates a decline in peroxisomal function with age, linking oxidative stress and disrupted lipid metabolism resulting from dysregulated peroxisomal function to various age-related diseases [17,18]. The accumulation of cholesterol is associated with peroxisomal dysfunction, and cholesterol oxide derivatives, known as oxysterols, are known to induce significant dysfunction in organelles, especially mitochondria. Oxysterols play pivotal roles in the disruption of redox homeostasis, inflammatory status, lipid metabolism, and ultimately, cell death induction during aging [19,20]. While specific phytosterols, also known as “plant cholesterol”, have been identified for their health benefits, recent research indicates that imbalances in these diet-derived phytosterols have substantial implications in neurodegeneration and cognitive decline [21]. Thus, antagonistic hallmarks seem to be beneficial to the cell when meticulously regulated but tend to become deleterious only at high levels. Integrative hallmarks arise when damage from primary and antagonistic aging accumulates, leading to stem cell exhaustion, altered intercellular communication, chronic inflammation, and dysbiosis [22,23,24]. Although these hallmarks appear to be independent entities, they are integrative, and a complex interplay of different regulatory pathways is inherent in driving the complex biological process of aging.

Earlier studies proposed that the accumulated burden of somatic postnatal mutations is the primary cause of aging [25,26,27]. As various theories on aging, particularly those centered on genetic changes within cells, evolved, a growing body of evidence also underscored the importance of epigenetic factors in the aging process [8,28,29]. While an organism’s genome maintains a relatively steady state throughout its lifetime, the epigenome undergoes extensive reprogramming. Thus, there is a growing interest in research challenging the prevailing idea that genetic aberrations primarily drive the aging process [30]. There are significant data to support the notion that normal cellular aging may occur despite fewer mutations, and conversely, cells with a higher mutation rate do not necessarily display premature aging [31,32,33]. The mounting evidence from yeast to humans indicates that the breakdown of epigenetic information, rather than genetic mutations, plays a pivotal role in aging [30,34,35,36]. The dysregulation of the cellular epigenome during aging and senescence is a complex phenomenon that manifests through various elements, including global histone levels, histone positioning on the DNA sequence, post-translational modifications (PTMs) of histones, histone variants, DNA methylation, and noncoding RNAs.

Eukaryotes have evolved a complex genome architecture composed of DNA and histones that synergistically regulate cell function and differentiation. The fundamental architecture of the chromatin, known as “beads-on-a-string”, refers to the nucleosome core particle. This nucleosome core comprises 147 base pairs of genomic DNA wrapped around a core histone octamer made of H2A, H2B, H3, and H4 [37]. The linker histone H1 stabilizes the chromatin by binding to specific sites on the DNA, anchoring the nucleosomes and facilitating the formation of higher-order chromatin structures. This compaction of chromatin around the core and linker histones is a key mechanism in eukaryotes to package and organize their genome efficiently. The higher-order chromatin structure can be classified as euchromatin and heterochromatin, depending on the level of compaction. Euchromatin corresponds to the less condensed regions of chromatin that allow for easy access to regulatory molecules such as transcription factors. On the other hand, heterochromatin represents transcriptionally inactive and highly condensed chromatin sites. The organization of these two distinct chromatin regions is closely associated with the histone content and organization of the nucleosome landscape. Nucleosomes are highly dynamic, both in their composition and positioning on the genome [38]. The genome is constantly reorganized through nucleosome eviction and histone sliding processes that are mediated by factors such as chromatin remodeling enzymes, histone chaperones, and polymerases [38,39,40]. Furthermore, nucleosomes in transcriptionally active or repressed sites carry multiple PTMs on histone proteins, which ultimately define the epigenomic state of the cell [41,42]. Among the vast repertoire of histone PTMs, acetylation, methylation, phosphorylation, and ubiquitination are the main modifications regulating the chromatin structure and gene transcription [41,43,44]. The accumulating evidence has shown that age-related dysregulation of the epigenome involves the breakdown of the nucleosome landscape, leading to changes in the organization of nuclear components, activation of previously silenced genes, and aberrant gene expression patterns [5,8].

Here, we will review recent findings that demonstrate the importance of histone expression profiles that occur in aging and examine how histone loss and nucleosome repositioning impact the chromatin architecture and numerous other critical homeostatic cellular processes. While some outstanding reviews in the last decade have shed light on histone modifications and their roles in aging [45,46,47,48], our emphasis is on exploring the breakdown of epigenetic information, specifically, histone loss, altered nucleosome occupancy, and the increased expression of noncanonical histone variants within the context of aging and senescence. Alterations in the abundance and composition of histones, along with histone PTMs, have been associated with changes in chromatin organization and gene expression patterns during aging. Histones, as key players in epigenetic regulation, have been implicated in the pathogenesis of various age-related diseases, including cancer and neurodegenerative disorders [49]. Evidence of aberrant histone modifications and changes in the chromatin structure is frequently observed in disease models characterized by premature aging, such as Hutchinson–Gilford progeria syndrome and Werner syndrome, emphasizing the relevance of histones in the context of age-related pathologies [50,51]. Gaining a deeper understanding of the intricate interplay between histone depletion and cellular aging is crucial for deciphering the underlying mechanisms that govern the fundamental aging-associated pathways and identifying potential therapeutic targets for various age-related pathologies.

## 2. Age-Related Histone Loss and Altered Nucleosome Occupancy in Non-Mammalian Models

For over 30 years, scientists have been studying global histone loss and its implications in cellular viability and aging in various yeast models. Early studies by Grunstein et al. provided compelling evidence that the depletion of histones H4 and H2B in yeast led to transcriptional defects, the disruption of the chromatin structure, and cell cycle arrest [52,53]. Subsequent investigations showed that the loss of histones H3 or H4 in yeast altered the transcriptional landscape with the derepression of many previously silenced genes [54,55]. Perhaps the most radical observation made by Hu et al. revealed an ~50% global depletion of histones and nucleosome repositioning in budding yeast during replicative aging. Histone removal from nucleosomes significantly enhanced DNA accessibility to transcriptional machinery, precipitating a sharp derepression of gene expression. Consequently, the yeast cells showed an overall upregulation in gene transcription and an increased genomic instability [56]. Similar to yeast studies, an age-dependent loss of histone H3 was reported in *Caenorhabditis elegans* when worm lysates were examined from young and old adults [57]. Thus, the mounting evidence from studies using yeast strains and other nonmammalian models via conditional histone knockouts or replicative aging has significantly enhanced our scientific understanding of the biologic effects of histone loss, altered nucleosome occupancy, differential gene expression, and cellular senescence (Table 1).

## 3. Histone Loss in Mammalian Models of Replicative and Chronological Aging

Mammalian cells progressively succumb to the adverse effects of aging assessed through two primary modalities: replicative life span and chronological life span [62,63]. Comprehensive investigations into these aging processes in yeast have demonstrated the emergence of aging hallmarks similar to those found in mammals [62,63,64]. Hence, evidence of the age-related effects on histone expression profiling of yeast cells captured the attention of researchers, prompting further investigations into histone levels in mammalian cells. An early study of histone depletion in human cells using pulse-chase labeling revealed a gradual decrease in H1 linker histones in senescent human diploid fibroblast cultures. This reduction was linked to a decline in the biosynthesis of H1 histone with progressive in vitro aging [65]. Houde and colleagues [66] provide a novel perspective on H1 expression changes in cultured human fibroblasts that exhibit a progressive cell cycle elongation. Their findings revealed that cells in early (28–35 mean population doubling; MPD) and late (65–70 MPD) passages, maintained in a confluent state at 0 and 6 weeks, exhibited a similar shift in the gene expression of H1 variants. Notably, this involved a reduction in H1B and a concurrent increase in H10 and H1A. These data suggest that changes in the expression levels of histone H1 variants occurred as a function of time of density-dependent growth rather than replicative age. Building on Mitsui et al.’s [65] discovery, subsequent studies focused on in vitro and in vivo aging in human skin fibroblasts and discovered a reduction in the nucleosome occupancy and an altered chromatin structure with aging [67,68]. While histone expression profiles in aging were being investigated in yeast, it took considerable time before Karsleder and colleagues delved into the correlation between histone levels and mammalian aging [69]. Their study using human diploid fibroblast (HDF) cell lines IMR90 and WI38 as models for replicative senescence revealed a significant decrease in histones H3 and H4 (43% and 47%, respectively) of late passage cells. The dysregulation of histone expression was primarily attributed to stress signals associated with telomere shortening. The telomeric region, affected by age-associated cellular damage, led to a DNA damage response and progressive genome-wide epigenetic changes with successive cell cycles that ultimately impacted histone biosynthesis. Chronic replicative stress only affected the H3 and H4 histones in HDFs, while the H1, H2A, and H2B histones remained unchanged [69]. Although the concept of histone loss in aging was initially suggested in actively replicating cells, Liu et al. studied the histone patterns of quiescent muscle stem cells (QSC) in young (2 months) and aged (24 months) mice and found a comparable decrease in the histone levels of H1, H2B, H3, and H4 [36]. This finding was quite significant in that histone loss also occurred during chronological aging in a quiescent cell type with minimal turnover.

While some aging models demonstrate a selective loss of canonical histones, recent research in mammals has revealed a global reduction in histone levels associated with aging. In activated naive CD4+ T cells from both young (20–35 years) and old (65–85 years) adults, a transcriptome analysis via RNA-seq showed that aged T cells tended to have globally reduced expressions of core histone genes compared to young cells, and this decrease was dependent on the cell cycle state of the activated, aged T cells. The global downregulation of histones, confirmed at both the mRNA and protein levels, was observed to delay the S-phase progression in aged T cells, thereby triggering a replication-stress response [70]. Similarly, a global loss of histones was observed in aged retinal pigment epithelium (RPE) cells, where all five canonical histones were depleted. To provide a comprehensive perspective, Dubey et al. assessed the histone expression in mouse RPE cells undergoing chronological aging and primary human RPE cell lines that reached senescence through replicative aging. In the RPE cells obtained from young (2–3 months) and aged (20–24 months) mice, a substantial reduction was observed in all core and linker histones. Furthermore, a replicative aging model using primary human RPE cells revealed a gradual decrease in the histone levels as the cells aged [71]. Together, these studies demonstrate that both actively proliferating and postmitotic cells in a quiescent state develop histone loss with aging. Also, a decreased histone expression in aged cells may be global or specific depending on the cell type and experimental model.

Much like aging, diverse senescence pathways can contribute to epigenetic dysregulation in cells [72]. Cellular senescence is an innate stress response, wherein cells enter a stable and irreversible state of cell cycle arrest even under optimal growth conditions. Senescent cells exhibit distinct changes in their morphology, chromatin structure, gene expression, and the emergence of a senescence-associated secretory phenotype (SASP), a chronic inflammatory state [73,74]. These alterations collectively cause the cell to exit the cell cycle permanently. Senescence is induced by multiple mechanisms including replicative senescence, senescence due to DNA damage, stress-induced senescence, and programmed developmental senescence [73,75]. In human RPE cells that attained replicative exhaustion and entered the senescent state, there was a substantial decrease in all canonical histones [71]. Furthermore, histone loss was observed in two different senescent models utilizing human umbilical vein endothelial cells (HUVECs) [76]. In the first model, senescence was induced through natural passaging until the cells reached a point of replicative exhaustion, while the second model involved prolonged exposure to TNF-alpha over 26 days. In both instances, senescent cells displayed a reduction in many histone isoforms along with the downregulation of genes responsible for regulating the cell cycle and DNA repair mechanisms. Regardless of the upstream stimuli prompting senescence, the process typically converges in downstream effects, with histone depletion emerging as one of the significant consequences. These findings underscore the importance of histone depletion as a critical factor driving the process of cellular senescence [76].

While the acquisition of senescence has been viewed as an alternative pathway to prevent cancer, the prolonged accumulation of senescent cells can paradoxically promote cancer development [77,78]. The emergence of SASP factors during senescence can impact the surrounding cells via alterations of the cellular microenvironments, establishing chronic inflammation and fostering conditions conducive to cancer [79]. Interestingly, histone depletion associated with aging and senescence has been inversely linked to malignancy, suggesting that histone loss can facilitate cell proliferation arrest, ultimately contributing to tumor suppression [80]. Unlike histone loss, histone mutations, known as oncohistones, are implicated in promoting cancers [81]. However, given that many age-dependent changes in the cellular epigenome resemble those observed in cancer, the epigenetic reprogramming occurring during aging may predispose individuals to cancer development [82]. Therefore, gaining a deeper understanding of age-related epigenomic changes holds the potential to elucidate the underlying causes of cancer.

## 4. Alterations in Nucleosome Landscape in Aging

The age-related loss of histone expression appears to be a widespread phenomenon observed in organisms ranging from yeast to mammals. However, there is limited research on how the distribution of nucleosomes across the genome changes as organisms age (Table 2). Celona et al. studied the impacts of histone loss and alterations in the nucleosome occupancy in yeast and mammalian cells. Their proposed model suggested that the loss of nucleosomes predominantly occurs in specific localized regions [60]. Mammalian cells with ablated high mobility group box 1 (HMGB1), a DNA-binding protein, showed a significant decrease in histones and nucleosomes. Their findings demonstrated that the shRNA- and siRNA-mediated ablation of HMGB1 in HeLa cells, or its yeast equivalent (Nhp6a/b), did not compromise the cell viability despite the histone loss. However, the HMGB1-deficient cells displayed a greater sensitivity to DNA damage, underwent a global increase in transcription, and showed specific alterations in the transcriptional profile of certain genes. This study proposed a chromatin model in which the loss of nucleosomes within the cells is not widespread, but rather localized, particularly in regions that inherently exhibit a lower tendency to form nucleosomes [60]. This model aligns with the genomic DNA code regulating nucleosome positioning, as postulated by Kaplan et al., and emphasizes the role of DNA sequence information in governing the arrangement of nucleosomes [83]. Thus, chromatin regions characterized by an innately lower tendency for histone deposition are more histone deficient. It is important to note that this study primarily aimed to unravel the fundamental genome organization under conditions of limited numbers of nucleosomes without delving into the aging aspect.

Similar to the findings of Celona and colleagues, a subsequent study demonstrated that rather than a global redistribution of nucleosomes, aging causes nucleosome loss at specific sites [84]. Bochkis et al. used high-throughput methods such as RNA-seq, MNase-seq, and ChIP-seq to map nucleosome changes in young (3 months) and aged (21 months) mouse livers. They discovered age-related differences in the nucleosome distribution, which in turn influenced the accessibility of chromatin to transcriptional factors. A reduction in the nucleosome density correlated with the elevated transcription of key genes involved in lipid metabolic pathways in aged livers, potentially leading to metabolic dysfunction and hepatic steatosis. Furthermore, there were specific sites with an increased nucleosome occupancy with age, including the serum response factor (*SRF*) gene that impacted its target genes involved in liver proliferation, lipid metabolism, and histone expression. The interrogation of histone isoforms revealed an irregular expression pattern displaying both upregulation and downregulation within aged hepatic tissue [84]. Although this study focused on the decline in SRF activity in the aging liver due to altered nucleosome occupancy, it is important to note that other mechanisms have been reported for repressed SRF activity including nuclear exclusion, protein kinase C-δ induced phosphorylation, and the inactivation of SRF in senescent cells [85,86]. In line with Bochkis et al.’s research, Chen and colleagues investigated H3 nucleosomal profiles in chronologically aged mouse tissues [87]. A comprehensive H3 nucleosomal map was generated using ChIP-Seq data from four tissues (heart, liver, cerebellum, and olfactory bulb) and one primary cell type (primary neural stem cells obtained from the subventricular zone) in mice from different age groups (3, 12, and 29 months). While they did not find significant changes in the global H3 levels, localized alterations in the H3 occupancy were observed in all the four tissue types and cultured neural stem cells during aging. These changes can manifest as an increased or decreased occupancy in specific regions with subsequent chromatin remodeling in aging tissues. Notably, the distal regions located 5–500 kb away from the annotated transcriptional start sites exhibited a significant remodeling of H3 occupancy, both upstream and downstream. The sites proximal to genes, particularly within intronic regions, consistently demonstrated robust nucleosome enrichment. Within aging chromatin, the distinct changes observed in distally located nucleosomes suggest a differential occupancy of nucleosomes in regulatory elements, particularly the forkhead transcription factors, which are critical regulators of DNA remodeling. Additionally, the repositioning of nucleosomes triggered transcriptional changes in inflammatory transcription factors such as STAT6 and IRF8. Together, these studies suggest that compared to the observations in actively replicating yeast and mammalian cells, histone depletion in chronologically aging tissues is relatively less and limited to specific genomic sites [87].

**Table 2 cells-13-00320-t002:** Histone depletion and nucleosome redistribution in human and other mammalian models.

Part B	Histone Alteration	Organism or Cell Line	Reason for Histone Alterations	Cellular Alterations	Reference
1	Histone H1 and histone H4	Human fetal lung fibroblast (TIG-1)	Replicative senescence	Age-related increase in nuclear proteins; decrease in histone H1 biosynthesis in senescent fibroblasts	Mitsui et al., 1980 [65]
2	Reduction in nucleosome occupancy	Human fibroblast	In vitro and in vivo aging of human diploid fibroblasts	Changes in chromatin structure	Ishimi et al., 1987 [67]
3	Reduction in nucleosome occupancy	Human donor fibroblast	In vitro and in vivo aging of human diploid fibroblasts	Changes in chromatin structure and cytoskeletal elements	Macieira-Coelho, 1991 [68]
4	Loss of histone H1	Human fibroblasts WI-38, MRC-5, IMR-90, and BJ	Cellular senescence induced by retroviral expression of oncogenic Ras (oncogene-induced senescence)	Exhibit unique chromatin condensation, termed senescence-associated heterochromatic foci (SAHF); loss of linker histone H1 accompanied by increased chromatin-bound high mobility group A2 (HMGA2) competitor protein	Funayama, 2006 [88]
5	Histone H3 and H4	Human diploid fibroblast IMR90 and WI38, fibroblast from young (9 y.o.) and 92 y.o. individual	Replicative aging in cell culture models and normal aging in primary cells	Downregulation of histone regulatory factors and histone chaperones (SLBP, Asf1a, Asf1b, CAF1-p150, and CAF1-p60); several altered histone modifications; accumulation of DNA damage and DNA damage response; telomere dysfunction	O’Sullivan et al., 2010 [69]
6	H1, H2A, H2B, H3, H4, and the variant histone H2AX	HeLa *Hmgb1−/−*	HMGB1 knockdown HeLa cells were generated using shRNA/siRNA	Increased global transcription and altered transcriptome profile in cells undergoing histone depletion	Celona et al., 2011 [60]
7	Altered histone expression and changes in nucleosome occupancy	Liver tissue from young (3 months) and old (21 months) C57BL6 mouse	Age-dependent changes	Age-related activation of lipogenesis and inflammatory genes; nucleosome occupancy changes with age selectively repress or derepress genes involved in lipid metabolism; histone isoforms are differentially regulated	Bochkis et al., 2014 [84]
8	HIST1H3D, HIST1H3E, and HIST4H4	CD8+ T cells from healthy young (22–40 yr) and old (65+ yr) individuals	Closed chromatin associated with aging	Age-associated changes to chromatin accessibility	Ucar et al., 2017 [89]
9	Altered histone H3 expression and changes in H3 nucleosome occupancy	Tissues (i.e., heart, liver, cerebellum, and olfactory bulb) and one primary cell type (i.e., primary neural stem cell cultures from the subventricular zone) from 3-, 12-, and 29-month-old mice	Chronological aging	Age-related alterations in nucleosome positioning exhibit localized changes in genomic loci linked to DNA remodeling factors; these changes also extend their influence to the regulation of inflammatory genes	Chen et al., 2020 [87]
10	H4 depletion and degradation of nucleosomes	Human fibroblasts IMR90, CRL-1474, and HEK-293T	Senescence-mediated H4 loss; proteasome-mediated H4 degradation	H4 and H3 concentrations are reduced at the promoter regions of cell cycle inhibitor genes, SASP-related genes, and anti-apoptotic genes	Lin et al., 2020 [90]
11	Global loss of core histones	Human naive CD4+ T cell from young (20–35 years) and old (65–85 years) adults	Age-related reduction of miR-181ab1 and consequent increase of its target histone deacetylase, SIRT1, leads to deacetylation of histone gene promoters and downregulation of histone genes	Replication stress, delayed S-phase progression, and activation of proinflammatory pathways	Kim et al., 2021 [70]
12	Histone reduction	Human fetal lung IMR90 diploid fibroblasts, neonatal human epidermal melanocytes	Senescent cells form cytoplasmic chromatin fragments (CCFs), and proteolysis of CCFs depletes total histones in a lysosome-dependent manner	Genomic DNA damage; deterioration of nuclear integrity; CCFs transition to the cytoplasm and subsequent histone loss from CCFs	Ivanov et al., 2013 [80]
13	H1, H2A, H2B, H3, and H4	Primary human umbilical vein endothelial cells (HUVECs)	Replicative senescence and inflammatory cytokine TNF-α-mediated senescence	Cell cycle arrest due to decline in cell cycle and mitosis regulatory factors; deterioration of DNA repair mechanisms; loss of chromatin architecture	Kandhaya-Pillai et al., 2023 [76]

## 5. Multiple Mechanisms of Histone Loss during Aging

Although aging is associated with histone loss and the reorganization of nucleosomes, the effects of histone depletion, nucleosome repositioning, and the factors driving histone reduction are still largely uncharted territories [4]. Studies in yeast offered critical insights into the genetic and epigenetic mechanisms behind the age-related dysregulation of histone expression. These mechanisms included telomere loss, DNA damage, changes in histone chaperones, reduced histone production, and alterations in histone marks, all of which affected the chromatin structure [91]. Despite limited research in mammals, similar causal links between decreasing histone levels and various genetic and epigenetic factors have been established in aging cells and tissues (Figure 1). For instance, the experiments on lung fibroblast cell lines undergoing replicative aging revealed that chronic exposure to DNA damage signals, such as those arising from telomere shortening, impacted histone biosynthesis. [69]. The presence of short telomeres and the consequent DNA damage signaling resulted in the depletion of the stem–loop binding protein (SLBP), a stabilizer of histone mRNA. This, in turn, reduced the biosynthesis of H3 and H4 and destabilized Asf1, a histone chaperone, leading to genome-wide epigenetic changes that amplified the damage signal during repeated cell cycles. This self-enforced regulatory loop constantly changed the chromatin environment at the telomeres, facilitating the incursion of damage markers beyond the threshold of cell tolerance, ultimately leading to irreversible cell cycle exit and senescence. The ectopic expression of telomerase enzymes in aging cells to increase the telomere length normalized the histone expression levels similar to those of young cells. Furthermore, this process partially restored SLBP and ASF1 expression, likely by alleviating the DNA damage signaling associated with short telomeres [69].

Telomere shortening and dysfunction are hallmarks of cellular aging and senescence. Despite their crucial role in maintaining chromosomal stability, the regulatory mechanisms governing telomeres during cellular aging remain poorly understood. Situated at the termini of linear chromosomes, telomeres form complexes of TTAGGG nucleotide repeats and proteins that regulate their functions, shielding them from recognition by the cell’s repair mechanism as double-stranded DNA breaks (DSBs). These proteins, including the telomerase (TERT) enzyme, histones, and the Shelterin complex, are critical for regulating the telomere length and preventing chromosomal end fusion [92,93].

Telomeres typically exhibit a heterochromatin structure, and a widespread phenomenon known as the telomere position effect results in low expression levels or the transcriptional silencing of genes within or near telomeres [94]. Additionally, telomeres tend to spatially organize at the nuclear periphery, a zone of transcriptional repression, in a cell cycle-dependent manner [95,96,97], and therefore, experience the transcriptional repression of genes. However, as cells approached senescence, a spatial overlap of lamina intranuclear structures with telomeres was observed [98]. During senescence, as the nuclear lamina’s organization gets disrupted, telomeres tend to form large aggregates lacking TERT. These telomere aggregates accumulated histone γ-H2AX, a classical marker of DSBs and telomere shortening, in senescent cells [98].

Furthermore, the formation of senescence-associated heterochromatin foci (SAHF), representing facultative heterochromatin domains, correlates with telomere shortening in cells entering senescence [99,100]. The SAHF contain domains with di- or tri-methylated lysine 9 of histone H3 (H3K9me2/3), a histone H2A variant (macroH2A), and heterochromatin protein 1 (HP1) proteins [100,101,102]. Additionally, epigenetic modifications such as histone methylation in the telomere region and TERT demethylation in humans play significant roles in maintaining the heterochromatin, transcriptional silencing at telomeres, and telomerase inactivation. Preserving the telomere structure and ensuring transcriptional silencing are critical to preventing premature aging [103,104].

Telomere shortening assumes that each successive cell division acts as a mitotic counting mechanism, eventually leading cells to attain replicative senescence [4,105]. In contrast, cells in a quiescent state transition into senescence despite negligible telomere shortening [106,107]. Therefore, the primary causes of age-associated histone loss in quiescent cells are likely driven by mechanisms beyond telomere dysfunction. Muscle stem cells provide a well-studied model for cellular quiescence and aging [108,109]. Liu and colleagues reported an epigenetic mechanism for histone loss in aging quiescent muscle stem cells (QSCs). Using the ChIP-Seq approach, the QSCs derived from young and aged mice were profiled for histone methylation marks, including H3K4me3, H3K27me3, and H3K36me3. As the QSCs underwent aging, there was an overall reduction in the expression of histone genes that acquired H3K27me3, a transcription repressive mark, at their transcription start sites. This finding underscores the critical role played by the repressive methylation mark in the regulation of histones within QSCs, simultaneously emphasizing the involvement of histone PTMs in histone regulation.

A similar methylation regulatory mechanism of histones was reported in fibroblast cell lines [90]. In this study, histone expression was examined in IMR90 fibroblast cells after inducing premature senescence via protein arginine methyltransferase 1 (*PRMT1*) knockdown. PRMT1 is the predominant arginine methyltransferase in humans, which mediates the asymmetric dimethylation of histone H4 at arginine 3 (H4R3me2as), a critical modification essential for histone H4 stability. Studies over the past decades have demonstrated the reduced expression of PRMT1 in replicative aging and senescent cells [110,111]. Lin et al. demonstrated that premature senescence in fibroblasts induced by *PRMT1* knockdown caused a reduction in PRMT1-mediated H4 dimethylation, leading to the destabilization of H4. Consequently, H4 showed increased binding to PA200, causing the ubiquitin-independent degradation of H4 by PA200-capped proteasomes [90,112] (described further in Section 7). A similar H4 degradation was also observed in fibroblasts exposed to different senescence-associated signals such as oxidative stress, DNA damage, or oncogenic signaling. Importantly, H4 degradation preceded the depletion of other histones, resulting in a reduced nucleosomal occupancy and impacted cell proliferation by enhancing the transcription of genes responsible for cell cycle inhibition, senescence-related genes, and apoptosis regulators [90]. It is critical to note that different histone marks have contrasting roles in regulating genes in the context of aging and lifespan across species and even within different tissues of an organism [47,113]. Therefore, recognizing the precise functions of various histone PTMs in aging tissues is critical for unraveling the complexity of epigenetic processes in aging.

Though a large part of the age-associated histone depletion in cells appears to stem from altered histone PTM marks, telomere dysfunction, and DNA damage, a recent study reported the miRNA-mediated downregulation of histones in aged T cells. Initially, Ucar et al. detected a significant compaction of chromatin in several histone genes (*HIST1H3D*, *HIST1H3E*, and *HIST4H4*) and histone modifiers in T cells with age [89]. By combining ATAC-seq and RNA-seq data, their study showed that immune cells undergo alterations in their epigenome as individuals age. This involves the closing of the chromatin structure at the promoter and enhancer regions of genes that are actively expressed, including histones. This finding was consistent with the reduced expression of core histones demonstrated in a subsequent study by Kim et al. using T cells from young (20–35 years) and old (65–85 years) adults [70]. Naive T cells from aged individuals with a reduced miR-181a level as well as miR-181ab1-deficient murine T cells in an actively dividing state showed the downregulation of histone expression, which induced replication stress and a consequential slow-down of the cell cycle. In the absence of miR-181a, its target molecule SIRT1—a histone deacetylase enzyme—binds directly to the histone promoter region. This interaction led to a reduction in histone acetylation, thereby inducing the repression of histone genes [70].

Although different studies provide insight into the ramifications of age-related genetic and epigenetic alterations of the nucleosome landscape, a key factor is that this mechanism is cell-type dependent. Interestingly, Ivanov and colleagues reported that histone loss in senescent cells occurred in both fibroblasts and epidermal melanocytes [80]. Cells approaching senescence and those already in a senescent state undergo significant stress, marked by changes in both the morphology and physiology. Genomic DNA damage, particularly DSBs, along with the deterioration of nuclear integrity in senescent cells, leads to damaged DNA fragments leaving the nucleus. Ivanov et al. identified these fragments in the cytoplasm as cytoplasmic chromatin fragments (CCFs), which are strongly positive for γ-H2AX and H3K27me3. In the cytoplasm, an autophagic proteolytic activity targets these CCFs, progressively depleting the total histones in a process dependent on the lysosomal activity [80]. This process of lysosomal ubiquitin-mediated histone proteolysis, elucidated by Ivanov and colleagues in senescent cells, represents an important mechanism of post-translational histone degradation. Together, these studies have enhanced our understanding of the intricate genetic and epigenetic pathways that control histone downregulation in the context of aging. Despite these insights, a consensus on the mechanism or function of histone loss during aging remains elusive, indicating the need for further research in this area.

## 6. Age-Associated Replacement of Canonical Isoforms by Histone Variants

Besides the canonical histones that mediate higher-order chromatin organization, mammals possess replication-independent, noncanonical variant histones (all canonical histones and their variants from humans are listed in Table 3). These variants contribute to creating distinct nucleosome structures and functions. Given the widespread distribution and multifaceted roles of histone variants, it is only natural that they exert their influence not only on normal physiological functions, but also on aging, senescence, and various pathogenic states. [114]. Many studies have revealed that replacing canonical histones with their variant counterparts impacts the chromatin dynamics and reprograms the transcriptome and epigenome of the cell [115,116]. One of the earliest studies reporting this phenomenon discovered the progressive accumulation of the H3.3 variant in rodent brains over a span of 150 days, with a concurrent decrease in the levels of canonical H3.1 and H3.2 histones [117]. H3.3, an evolutionarily highly conserved variant, has been increasingly recognized for its role in regulating developmental processes within organisms [118,119,120]. The depletion of H3.3 had detrimental effects on embryonic *Mus musculus* and *Xenopus*, underlining its critical role in maintaining the genome integrity during tissue maturation [118,120,121,122,123]. The interaction of the H3.3 variant with histone chaperone complexes like HIRA and ATRX/DAXX facilitated its incorporation at important chromatin regions such as protein-coding genes, telomeres, and pericentric chromatin. The HIRA complex-mediated deposition of the H3.3 variant in HeLa cells was identified as a potential cellular repair mechanism to fill DNA regions lacking nucleosomes, thereby reinstating nucleosomal organization [124].

Several reports demonstrated that H3.3 accumulated in aged and senescent cells, effectively replacing the H3.1 and H3.2 canonical histones. In the somatic tissues of mice, including the liver, kidney, heart, and brain, histone H3.3 progressively replaces the canonical histones, achieving near-saturation levels of neuronal nucleosomes by mid-adolescence [125,126,127]. A similar pattern was observed in the human brain, where H3.3 constituted >93% of the total H3 pool across individuals aged 14 to 72 years [125]. The accumulation of H3.3 was accompanied by an altered methylation landscape of tissue, reflecting the epigenetic plasticity in response to the association of different histone isoforms with the genome [126,127]. Furthermore, in fibroblasts and melanocytes, the proteolytic cleavage of H3.3 led to the integration of its cleaved product, known as H3.3cs1, into the nucleosome structure. Both H3.3 and its derivative H3.3cs1 induced senescence by suppressing genes linked to cell proliferation [128].

With the vast repertoire of H2A variants expressed in mammalian cells, they mainly account for the nucleosome diversity [129]. Histone H2A has functionally distinct variants like H2A.B, H2A.X, H2A.Z, and macroH2A that play a significant role in maintaining the nucleosome. Studies demonstrating the accumulation of H2A variants, macroH2A, H2A.Z, and γ-H2AX in the context of aging and senescence revealed the impact these variants have on the transcription and regulation of senescence biomarkers, including SASP genes [130,131,132]. H2A.Z, a promoter of heterochromatin, is increased in the hippocampi of aging mice. Conversely, the removal of H2A.Z from nucleosomes enhanced the gene expression of learning-associated genes [130]. Senescent cells, showing γ-H2AX accumulation as a result of persistent DNA damage, were found to regulate p53-dependent senescent growth arrest and senescence-associated extracellular inflammatory signaling [132,133]. Senescent fibroblast cells harbored elevated levels of the H2B type 1-K variant, but the accumulation was exclusive to cells with persistent DNA damage [134]. Thus, DNA damage-associated variant histone exchange within senescent cells likely serves to maintain the genome integrity by nucleosome deposition in damaged chromatin regions. The histone H2A variant, H2A.J, also accumulated in fibroblasts undergoing replicative senescence and supplanted canonical H2A expression in cells with persistent DNA damage [135,136]. The H2A.J variant was found to regulate inflammatory gene expression, as the depletion or overexpression of H2A.J regulated essential SASP genes either negatively or positively, respectively [136]. Similarly, a gradual buildup of the H2A.J variant was observed in epidermal keratinocytes from donors between 18–90 years, correlating to their chronological age [137]. The H1 histones are a diverse class comprising eleven variants in mammals [138,139], and of these, H1.0, a replication-independent variant, is increased in terminally differentiated cells [140]. Several reports over the last decade have highlighted the importance of H1.0 and other H1 variants in the regulation of the heterochromatin structure and genome remodeling.

These studies illustrate that the replacement of canonical histones by variants with age occurs in a cell-type-specific or context-specific manner [141]. In various cellular processes, studies have shown that certain histone variants possess unique functions and are vital for the survival of particular organisms [142,143]. Mutations in these variants can lead to severe defects, although they might be nonessential for other organisms [142,143,144,145]. The regulated deposition of histone variants plays a crucial role in preserving the senescent state of the cell, acting as a barrier against the development of cancer [128,146]. Mutations in these histone variants can result in severe defects and are commonly observed in association with various cancers [146]. The intricate interplay between these histone variants in senescent cells underscores their significance in maintaining genomic integrity, preventing uncontrolled cell proliferation, and potentially transforming into cancerous cells. Additionally, the sequence of the genome and the organization of chromatin influence the pattern of incorporation and tissue specificity of histone variants [147,148]. Hence, in the aging process, biological factors such as DNA damage, SASP activation, and the expression of inflammation genes likely contribute to the integration of specific variants into the nucleosome (Figure 2).

**Table 3 cells-13-00320-t003:** List of human histones and histone variants [149].

Human H1 Histones				
Variant Symbol	Previous Symbol	HGNC Gene Symbol	HGNC ID	HGNC Gene Name
H1.0	H1F0	*H1-0*	HGNC:4714	H1.0 linker histone
H1.1	HIST1H1A	*H1-1*	HGNC:4715	H1.1 linker histone, cluster member
H1.2	HIST1H1C	*H1-2*	HGNC:4716	H1.2 linker histone, cluster member
H1.3	HIST1H1D	*H1-3*	HGNC:4717	H1.3 linker histone, cluster member
H1.4	HIST1H1E	*H1-4*	HGNC:4718	H1.4 linker histone, cluster member
H1.5	HIST1H1B	*H1-5*	HGNC:4719	H1.5 linker histone, cluster member
H1.6	HIST1H1T	*H1-6*	HGNC:4720	H1.6 linker histone, cluster member
H1.7	H1FNT	*H1-7*	HGNC:24893	H1.7 linker histone
H1.8	H1FOO	*H1-8*	HGNC:18463	H1.8 linker histone
NA	HILS1	*H1-9P*	HGNC:30616	H1.9 linker histone, pseudogene
H1.10	H1FX	*H1-10*	HGNC:4722	H1.10 linker histone
NA	HIST1H1PS1	*H1-12P*	HGNC:19163	H1.12 linker histone, cluster member pseudogene
Human H2A Histones				
Variant Symbol	Previous Symbol	HGNC Gene Symbol	HGNC ID	HGNC Gene Name
H2A	HIST1H2AA	*H2AC1*	HGNC:18729	H2A clustered histone 1
NA	HIST1H2APS1	*H2AC2P*	HGNC:18720	H2A clustered histone 2, pseudogene
NA	HIST1H2APS2	*H2AC3P*	HGNC:18804	H2A clustered histone 3, pseudogene
H2A	HIST1H2AB	*H2AC4*	HGNC:4734	H2A clustered histone 4
NA	HIST1H2APS5	*H2AC5P*	HGNC:4728	H2A clustered histone 5, pseudogene
H2A	HIST1H2AC	*H2AC6*	HGNC:4733	H2A clustered histone 6
H2A	HIST1H2AD	*H2AC7*	HGNC:4729	H2A clustered histone 7
H2A	HIST1H2AE	*H2AC8*	HGNC:4724	H2A clustered histone 8
NA	HIST1H2APS3	*H2AC9P*	HGNC:18805	H2A clustered histone 9, pseudogene
NA	HIST1H2APS4	*H2AC10P*	HGNC:4732	H2A clustered histone 10, pseudogene
H2A	HIST1H2AG	*H2AC11*	HGNC:4737	H2A clustered histone 11
H2A	HIST1H2AH	*H2AC12*	HGNC:13671	H2A clustered histone 12
H2A	HIST1H2AI	*H2AC13*	HGNC:4725	H2A clustered histone 13
H2A	HIST1H2AJ	*H2AC14*	HGNC:4727	H2A clustered histone 14
H2A	HIST1H2AK	*H2AC15*	HGNC:4726	H2A clustered histone 15
H2A	HIST1H2AL	*H2AC16*	HGNC:4730	H2A clustered histone 16
H2A	HIST1H2AM	*H2AC17*	HGNC:4735	H2A clustered histone 17
H2A	HIST2H2AA3	*H2AC18*	HGNC:4736	H2A clustered histone 18
H2A	HIST2H2AA4	*H2AC19*	HGNC:29668	H2A clustered histone 19
H2A	HIST2H2AC	*H2AC20*	HGNC:4738	H2A clustered histone 20
H2A	HIST2H2AB	*H2AC21*	HGNC:20508	H2A clustered histone 21
H2A	HIST3H2A	*H2AC25*	HGNC:20507	H2A clustered histone 25
H2A.Z.1	H2AFZ	*H2AZ1*	HGNC:4741	H2A.Z variant histone 1
H2A.Z.2	H2AFV	*H2AZ2*	HGNC:20664	H2A.Z variant histone 2
macroH2A.1	H2AFY	*MACROH2A1*	HGNC:4740	macroH2A.1 histone
macroH2A.2	H2AFY2	*MACROH2A2*	HGNC:14453	macroH2A.2 histone
H2A.X	H2AFX	*H2AX*	HGNC:4739	H2A.X variant histone
H2A.J	H2AFJ	*H2AJ*	HGNC:14456	H2A.J histone
H2A.B	H2AFB1	*H2AB1*	HGNC:22516	H2A.B variant histone 1
H2A.B	H2AFB2	*H2AB2*	HGNC:18298	H2A.B variant histone 2
H2A.B	H2AFB3	*H2AB3*	HGNC:14455	H2A.B variant histone 3
H2A.P	HYPM	*H2AP*	HGNC:18417	H2A.P histone
NA	NA	*H2AQ1P*	HGNC:53962	H2A.Q variant histone 1, pseudogene
H2A.L	NA	*H2AL1Q*	HGNC:53959	H2A.L variant histone 1Q
NA	NA	*H2AL1MP*	HGNC:53961	H2A.L variant histone 1 M, pseudogene
H2A.L	NA	*H2AL3*	HGNC:53960	H2A.L variant histone 3
Human H2B Histones				
Variant Symbol	Previous Symbol	HGNC Gene Symbol	HGNC ID	HGNC Gene Name
H2B	HIST1H2BA	*H2BC1*	HGNC:18730	H2B clustered histone 1
NA	HIST1H2BPS1	*H2BC2P*	HGNC:18719	H2B clustered histone 2, pseudogene
H2B	HIST1H2BB	*H2BC3*	HGNC:4751	H2B clustered histone 3
H2B	HIST1H2BC	*H2BC4*	HGNC:4757	H2B clustered histone 4
H2B	HIST1H2BD	*H2BC5*	HGNC:4747	H2B clustered histone 5
H2B	HIST1H2BE	*H2BC6*	HGNC:4753	H2B clustered histone 6
H2B	HIST1H2BF	*H2BC7*	HGNC:4752	H2B clustered histone 7
H2B	HIST1H2BG	*H2BC8*	HGNC:4746	H2B clustered histone 8
H2B	HIST1H2BH	*H2BC9*	HGNC:4755	H2B clustered histone 9
H2B	HIST1H2BI	*H2BC10*	HGNC:4756	H2B clustered histone 10
H2B	HIST1H2BJ	*H2BC11*	HGNC:4761	H2B clustered histone 11
H2B	HIST1H2BK	*H2BC12*	HGNC:13954	H2B clustered histone 12
H2B	HIST1H2BL	*H2BC13*	HGNC:4748	H2B clustered histone 13
H2B	HIST1H2BM	*H2BC14*	HGNC:4750	H2B clustered histone 14
H2B	HIST1H2BN	*H2BC15*	HGNC:4749	H2B clustered histone 15
NA	HIST1H2BPS2	*H2BC16P*	HGNC:4754	H2B clustered histone 16, pseudogene
H2B	HIST1H2BO	*H2BC17*	HGNC:4758	H2B clustered histone 17
H2B	HIST2H2BF	*H2BC18*	HGNC:24700	H2B clustered histone 18
NA	HIST2H2BD	*H2BC19P*	HGNC:20517	H2B clustered histone 19, pseudogene
NA	HIST2H2BC	*H2BC20P*	HGNC:20516	H2B clustered histone 20, pseudogene
H2B	HIST2H2BE	*H2BC21*	HGNC:4760	H2B clustered histone 21
H2B	HIST3H2BB	*H2BC26*	HGNC:20514	H2B clustered histone 26
NA	HIST3H2BA	*H2BC27P*	HGNC:20515	H2B clustered histone 27, pseudogene
H2B.K	H2BE1	*H2BK1*	HGNC:53833	H2B.K variant histone 1
NA	H2BP4	*H2BL1P*	HGNC:54442	H2B.L histone variant 1, pseudogene
H2B.W	H2BFWT	*H2BW1*	HGNC:27252	H2B.W histone 1
H2B.W	H2BFM	*H2BW2*	HGNC:27867	H2B.W histone 2
NA	NA	*H2BW3P*	HGNC:44390	H2B.W histone 3, pseudogene
NA	H2BFXP	*H2BW4P*	HGNC:25757	H2B.W histone 4, pseudogene
H2B.N	NA	*H2BN1*	HGNC:56200	H2B.N variant histone 1
H2B	H2BFS	*H2BC12L*	HGNC:4762	H2B clustered histone 12 like
Human H3 Histones				
Variant Symbol	Previous Symbol	HGNC Gene Symbol	HGNC ID	HGNC Gene Name
H3.1	HIST1H3A	*H3C1*	HGNC:4766	H3 clustered histone 1
H3.1	HIST1H3B	*H3C2*	HGNC:4776	H3 clustered histone 2
H3.1	HIST1H3C	*H3C3*	HGNC:4768	H3 clustered histone 3
H3.1	HIST1H3D	*H3C4*	HGNC:4767	H3 clustered histone 4
NA	NA	*H3C5P*	HGNC:54427	H3 clustered histone 5, pseudogene
H3.1	HIST1H3E	*H3C6*	HGNC:4769	H3 clustered histone 6
H3.1	HIST1H3F	*H3C7*	HGNC:4773	H3 clustered histone 7
H3.1	HIST1H3G	*H3C8*	HGNC:4772	H3 clustered histone 8
NA	HIST1H3PS1	*H3C9P*	HGNC:18982	H3 clustered histone 9, pseudogene
H3.1	HIST1H3H	*H3C10*	HGNC:4775	H3 clustered histone 10
H3.1	HIST1H3I	*H3C11*	HGNC:4771	H3 clustered histone 11
H3.1	HIST1H3J	*H3C12*	HGNC:4774	H3 clustered histone 12
H3.2	HIST2H3D	*H3C13*	HGNC:25311	H3 clustered histone 13
H3.2	HIST2H3C	*H3C14*	HGNC:20503	H3 clustered histone 14
H3.2	HIST2H3A	*H3C15*	HGNC:20505	H3 clustered histone 15
H3.3	H3F3, H3F3A	*H3-3A*	HGNC:4764	H3.3 histone A
H3.3	H3F3B	*H3-3B*	HGNC:4765	H3.3 histone B
H3.4	HIST3H3	*H3-4*	HGNC:4778	H3.4 histone, cluster member
H3.5	H3F3C	*H3-5*	HGNC:33164	H3.5 histone
NA (H3.6)	H3F3AP6	*H3P16*	HGNC:42982	H3 histone pseudogene 16
H3.7	HIST2H3PS2	*H3-7*	HGNC:32060	H3.7 histone (putative)
NA (H3.8)	H3F3AP5	*H3P44*	HGNC:42981	H3 histone pseudogene 44
H3.Y.1	NA	*H3Y1*	HGNC:43735	H3.Y histone 1
H3.Y.2	NA	*H3Y2*	HGNC:43734	H3.Y histone 2
cenH3	NA	*CENPA*	HGNC:1851	Centromere protein A
Human H4 Histones				
Variant Symbol	Previous Symbol	HGNC Gene Symbol	HGNC ID	HGNC Gene Name
H4	HIST1H4A	*H4C1*	HGNC:4781	H4 clustered histone 1
H4	HIST1H4B	*H4C2*	HGNC:4789	H4 clustered histone 2
H4	HIST1H4C	*H4C3*	HGNC:4787	H4 clustered histone 3
H4	HIST1H4D	*H4C4*	HGNC:4782	H4 clustered histone 4
H4	HIST1H4E	*H4C5*	HGNC:4790	H4 clustered histone 5
H4	HIST1H4F	*H4C6*	HGNC:4783	H4 clustered histone 6
H4	HIST1H4G	*H4C7*	HGNC:4792	H4 clustered histone 7
H4	HIST1H4H	*H4C8*	HGNC:4788	H4 clustered histone 8
H4	HIST1H4I	*H4C9*	HGNC:4793	H4 clustered histone 9
NA	HIST1H4PS1	*H4C10P*	HGNC:4786	H4 clustered histone 10, pseudogene
H4	HIST1H4J	*H4C11*	HGNC:4785	H4 clustered histone 11
H4	HIST1H4K	*H4C12*	HGNC:4784	H4 clustered histone 12
H4	HIST1H4L	*H4C13*	HGNC:4791	H4 clustered histone 13
H4	HIST2H4A	*H4C14*	HGNC:4794	H4 clustered histone 14
H4	HIST2H4B	*H4C15*	HGNC:29607	H4 clustered histone 15
H4	HIST4H4	*H4C16*	HGNC:20510	H4 histone 16

## 7. Multilevel Regulation of Histone Degradation

Cells have developed robust mechanisms to synthesize core and linker histones abundantly during the S-phase of the cell cycle, ensuring a sufficient supply during chromatin duplication [150,151,152,153]. The regulation of histone biosynthesis occurs at various levels encompassing histone gene transcription, mRNA stability, translation efficiency, and a wide array of histone PTMs [154,155]. Extensive reviews have delved into these processes that have significantly enhanced our understanding of histone metabolism and its correlation to cell cycle checkpoints [156,157]. Following synthesis, histone density is regulated at both the post-transcriptional and post-translational levels, and aging studies have demonstrated the vital roles these regulatory mechanisms play in histone depletion [36,70,90]. Here, we will explore the final frontier of histone regulation, mainly focusing on histone degradation at both the mRNA and protein levels. This phenomenon is observed under specific cellular conditions such as extensive chromatin remodeling, the formation of relaxed chromatin structures with increased access to DNA, DNA damage response, and a surplus of histones. Histone degradation, documented across various developmental stages such as spermiogenesis, embryogenesis, senescence, and aging, plays a pivotal role in meticulously controlling the histone pool. This regulation is vital for maintaining the equilibrium required in the dynamic processes of chromatin assembly and disassembly.

### 7.1. Histone mRNA Degradation

In mammalian cells, there is a substantial increase (~35-fold) in histone transcription at the G1/S transition phase of the cell cycle, followed by rapid histone mRNA degradation [151,158,159]. Typically, histone mRNAs lack poly-A tails and instead have a conserved 25–26 nucleotide sequence at their 3′ ends, forming a unique stem–loop structure crucial for swift mRNA decay at the end of the S-phase (Figure 3). The stem–loop binding protein (SLBP) specifically binds to the stem–loop structure and plays a central role in the post-transcriptional processing, translation, and degradation of histone mRNAs [156,160,161]. Cells employ a potent feedback inhibition mechanism to regulate their histone pool. Soluble histones are central to this mechanism, as they enable the degradation of histone mRNAs, coinciding with the inhibition of DNA synthesis [158,162,163]. Initially, the nuclear autoantigenic sperm protein (NASP), a histone chaperone, shields these soluble histones from degradation, protecting them right from their synthesis until they are integrated into the nucleosome unit [164,165]. However, an excess of soluble histones can disturb the SLBP-3’ stem–loop interaction, ultimately resulting in mRNA degradation through the action of the exonuclease 3′hExo [166,167].

Similar to histones, SLBP also follows a cell cycle-regulated pattern of expression. SLBP interacts with phosphorylated upstream frameshift 1 (UPF1), an important RNA decay factor [168]. This interaction forms the SLBP–UPF1 complex, which recruits SMG5 and PNRC2 (proline-rich nuclear receptor coregulatory protein 2), initiating histone mRNA decapping by DCP1A/2 enzymes [169]. Subsequently, degradation occurs through a bidirectional 5′-to-3′ XRN1-mediated exoribonucleolytic activity and 3′-to-5′ exosome degradation [166,170,171]. A prerequisite for histone mRNA decay is the TUTase enzyme-, TUT7-, and/or TUT4-mediated modification of the 3′ end by oligourydilation [166,170]. The TUT enzymes provide a scaffold for the Lsm 1–7 heptamer complex, which along with SLBP, coordinates histone mRNA degradation through a distinct set of nucleases [172,173]. In mammals, these decay mechanisms for histone mRNAs can occur simultaneously or independently to regulate histones.

### 7.2. Histone Protein Degradation

The abundance of histones in a cell is also regulated through proteolytic degradation, a process facilitated by multiple pathways such as through proteases, the autophagy–lysosomal system, and the proteasome complex (Figure 4). The histone proteins integrated into nucleosomes exhibit a remarkable stability, with half-lives ranging from several days to months, but this is significantly influenced by cellular proliferation and the differentiation status [174,175]. Nevertheless, soluble histones not bound to chromatin undergo rapid degradation at the end of the S-phase. Therefore, the maintenance of homeostasis in the histone pool results from the synergistic degradation of free and chromatin-bound histones through the proteasomal and autophagy–lysosomal systems.

Histones are channeled through both ubiquitin-dependent and ubiquitin-independent pathways for proteasome-mediated degradation [176]. Ubiquitination constitutes an important histone PTM wherein a highly conserved, 76-amino-acid ubiquitin tag typically conjugates to lysine residues of the target histone proteins [177]. This modification serves as a signal for protein degradation through the proteasome complex [178,179]. The ubiquitination process involves a coordinated cascade of enzymes, including E1 ubiquitin-activating enzymes, E2 ubiquitin-conjugating enzymes, and E3 ubiquitin ligases [180]. Sequential rounds of ubiquitination on the lysine residues of the initial histone-conjugated ubiquitin result in the polyubiquitination of histone substrates, thereby facilitating the binding of polyubiquitin chains to the ubiquitin receptor on the proteasome complex [181,182]. The proteasome itself is a large, hollow, multisubunit complex that exists in two primary forms within cells: the 20S and 26S species. The 20S proteasome is a four-stacked ring structure and houses six active protease sites within its central cavity, making it the catalytic core of this complex. The openings to this cavity exclusively admit denatured proteins that are systematically degraded into smaller peptides. Various regulatory particles (RP) are positioned at one or both ends of the 20S proteasome, carrying out diverse functions including substrate recognition and their subsequent translocation to the catalytic core [183,184]. A well-known RP, 19S, complexes with the 20S core to form the 26S proteasome. Depending on their intracellular localization, different RPs such as PA28γ, PA200 (primarily located in the nucleus), and PA28α and β (predominantly cytosolic) associate with the 20S core and assume critical roles in facilitating protein deubiquitination, denaturation, and translocation into the 20S proteasomal core, eventually leading to degradation. Histone degradation by ubiquitin tagging is known to occur in various cellular contexts. Histone turnover is a critical mechanism that regulates DNA accessibility during transcription and replication, DNA damage repair, and the excess accumulation of histones [185]. When histones undergo ubiquitin-dependent degradation, it is often accompanied by the addition or removal of histone PTM marks, revealing an intricate link between the ubiquitin-dependent degradation process and the regulation of epigenetic marks on the histones. Studies have shown that denaturation and deubiquitination are prerequisites for a polyubiquitinated protein’s entry into the proteasome core for degradation [186]. However, the proteasome-mediated degradation of denatured proteins can also occur independently of the ubiquitin tag. This ubiquitin-independent degradation of histones is observed in response to DNA damage and spermiogenesis [112,187]. The 20S proteasome core, associated with RP PA200, specifically degrades the core histones bearing specific PTM marks [112].

Cells also utilize the autophagy pathway to degrade excess histones, safeguarding themselves from potential detrimental effects. Chaperone-mediated autophagy (CMA) has been identified as the primary pathway responsible for maintaining optimal histone levels, specifically by breaking down newly synthesized H3 and H4 histones [188]. During the maturation process of newly formed histones, a series of steps ensure their proper folding and dimerization. In mammals, the NASP chaperone plays a crucial role in regulating the soluble histone pool of H3 and H4. NASP is a cell cycle-regulated protein that shows a bidirectional modulation of the histone reservoir. A precisely regulated interaction between NASP and its cochaperones—HSC70 (for H3) and HSP70/HSP90/PP32/Set (for H4)—targets histones for lysosomal degradation through the lysosomal surface receptor protein LAMP2A [188,189].

## 8. Histone Complementation in Cells Experiencing Histone Loss

Budding yeast is a versatile organism well-suited for studying eukaryotic aging due to its finite lifespan and genetic manipulability. Studies using yeast models have shown that as cells age, they experience a significant loss of histones. Feser and colleagues demonstrated that elevating the histone expression in aging yeast cells extended their lifespan effectively [59]. Mutant yeast strains lacking specific histone chaperone genes like asf1, rtt109, and hir1 exhibited distinct histone mRNA levels. Short-lived mutants (asf1 and rtt109) displayed diminished histone levels, while hir1 mutants, marked by increased histone levels, exhibited prolonged lifespans. Furthermore, the overexpression of histone genes using a high copy number plasmid in both wild-type and yeast mutants with a shorter lifespan resulted in an increased cellular longevity. Although this discovery seemed promising for addressing histone loss in aging and senescent cells, a subsequent yeast study highlighted the adverse effects, including genomic instability and cytotoxicity associated with histone overexpression [190]. Persistent histone production beyond the S-phase resulted in aberrant cell division, defective chromosome segregation, whole genome duplications, and reduced histone variant incorporation into the chromatin [191,192,193]. Further research in yeast by Yu and colleagues demonstrated an intriguing correlation between histone dosage and replicative lifespan. It was observed that histone biosynthesis surpassed requisite levels in young cells but gradually declined with age [194]. Despite these insights into yeast, it remains unclear whether histone overexpression promotes longevity and addresses aging challenges in mammalian cells. Understanding the cellular sensitivity to histone levels and the consequences of aberrant histone expression in mammalian models becomes crucial in light of the contrasting effects observed in yeast.

Histones are basic proteins and, in excess, can disrupt cellular functions, causing defects in mitotic chromosome segregation through their interaction with negatively charged molecules [195,196]. Therefore, maintaining an optimal histone supply is crucial for cellular homeostasis. Histone biosynthesis is tightly coordinated with DNA replication, involving a wide array of regulatory molecules to maintain a nucleosomal balance in actively proliferating or quiescent cells [155,197]. In a recent study, Kim et al. demonstrated that aged T cells exhibited reduced histone levels due to the elevated levels of SIRT1, a histone deacetylase [70]. Inhibiting SIRT1 in aged T cells resulted in an increased histone gene expression, restored the cell cycle progression, and mitigated the replication-stress response. Additionally, an in vivo mouse model of T-cell aging revealed an enhanced T-cell activation during the murine antiviral response upon SIRT1 inhibition. Thus, modulating the SIRT1 activity to replenish histone levels appeared to have a positive impact by restoring cellular functionality. In a subsequent study by Yang et al. the authors used a transgenic mouse model called ‘inducible changes to the epigenome’ (ICE) that expresses I-PpoI endonuclease in a tamoxifen-inducible manner. The ICE mice allowed for the introduction of DSBs in 19 noncoding regions in the genome with no evidence of mutations or immediate deleterious effects. However, at ten months postinduction, the ICE mice displayed aging characteristics accompanied by alterations in epigenetic patterns and transcriptional activity at the histone level. Using this model, the researchers effectively demonstrated that by inducing the expression of Yamanaka factors (OSK), they could successfully reprogram the epigenetic profile to resemble a more youthful state, thereby restoring the normal histone expression [30]. Together, these studies showed that certain regulatory factors play a central role in the expression of histone genes, and tweaking the expression of these master regulators could attenuate or accentuate the histone levels [60,198,199,200,201].

Recent research in mammalian models demonstrated the restoration of histone expression via treatment with pharmacological compounds, including rapamycin, metformin, and resveratrol. These drugs are increasingly recognized for their geroprotective and anti-aging properties, as evidenced by clinical studies demonstrating improved outcomes in age-related pathologies [202]. In a recent study by Lu et al., a robust upregulation of H3 and H4 histones was observed with the administration of rapamycin, an inhibitor of the protein kinase mTORC1 [203]. The study found that rapamycin induced an increased histone expression in the intestinal enterocytes of drosophila and also alleviated the age-associated decline in a mouse model by promoting gut health and homeostasis. Rapamycin-treated enterocytes exhibited a chromosomal rearrangement, redistribution of heterochromatin domains, and activation of autophagy. Although the primary focus of this study was on the effects of rapamycin-induced histone upregulation in Drosophila, it also highlighted similar outcomes in mouse intestinal autophagy, gut health, and lifespan. The restoration of histones using rapamycin in this study identified the mTORC1–histone axis as a critical prolongevity mechanism in mammals.

Another study comparing the transcriptome of muscle progenitor cells (MPC) derived from old and young participants showed the reduced expression of several histone isoforms [204]. However, in older participants who ingested metformin for two weeks, there was partial restoration of histone levels and improved function of myoblasts. Although this study did not delve into the precise cellular mechanisms reversing histone loss, metformin ingestion altered the transcriptional profile of old MPCs, significantly upregulating the histone genes. Similarly, in IMR90 fibroblast cells experiencing H4 histone loss due to H_2_O_2_-induced senescence, the restoration of H4 and delayed onset of senescence were observed within 24 h of treatment with rapamycin, metformin, and resveratrol [90]. While this study did not elaborate on the mechanism of H4 renewal, previous reports suggested that rapamycin prevented the degradation of ubiquitinylated substrates like histones by binding to the proteasome subunit. Collectively, these findings suggest that the response of histone genes to these anti-aging drugs could be effective in regulating molecules that are at the crossroads of aging and age-related histone loss.

## 9. Conclusions and Future Perspectives

The loss of histones and changes in nucleosome occupancy, among other classical epigenomic biomarkers like the global decline in genomic DNA methylation, heterochromatin loss, altered patterns of histone PTMs, and deregulation of noncoding RNAs are recognized as hallmarks of mammalian aging and senescence [4]. This review provides a comprehensive overview of the changing histone profile and nucleosome occupancy in the aging cell epigenome, drawing insights from human, animal, and cell-based models. Systematic studies in yeast have demonstrated the functional significance of histone loss and restoration in cellular aging and senescence. However, unraveling the mechanisms behind histone loss in aging mammals, especially humans, poses technical challenges due to the complex encoding of histones by various gene clusters distributed across multiple chromosomal locations. As an added layer of regulatory control, histone expression is a multitiered process involving numerous transcription factors and chaperones. Furthermore, the redistribution and histone profiles of nucleosomes exert varied effects on cellular aging and longevity through the differential regulation of the transcriptome and epigenomic landscape across species, and even within tissues of the same species. Despite these challenges, it is critical to determine (1) potential cellular mechanisms maintaining histone levels and identify key factors contributing to age-related histone depletion and genetic instability and (2) whether changes in the histone expression could be leveraged to mitigate the effects of aging and extend the lifespans of multicellular organisms.

Several recent technological advances offer promising avenues for addressing these challenges. Recent advances in genome editing, like the CRISPR–Cas9 technology, open the possibility of a systematic, targeted manipulation of histone regulatory genes in mammalian systems to determine the functional requirements of various regulatory factors. While the use of geroprotective drugs like metformin, rapamycin, and resveratrol presents a promising approach for rejuvenating aging cells experiencing histone depletion in model organisms, the critical question remains whether these drugs can achieve sufficient selectivity among different histone variants and PTMs to effectively address age-related pathologies. A substantial gap in our understanding pertains to the efficacy of various geroprotectors across diverse human tissues. Existing data are largely derived from a limited number of studies, predominantly utilizing mouse models or in vitro cell cultures, which may not accurately reflect the pathological conditions in humans [205,206]. The targeting of epigenetic factors and markers of senescence has been at the core of several clinical trials, but precise clinical trials targeting histones or histone regulatory molecules have not been performed yet (NCT03353597, NCT00242255, NCT06065241, and NCT05392582). An evaluation of the clinical efficacy of these geroprotective drugs is crucial, as Juricic et al. recently demonstrated in Drosophila that an increased histone expression is due to the “memory effect” of chronic rapamycin treatment [207]. Rapamycin is known for its ability to inhibit the mechanistic target of rapamycin (mTOR) pathway, which is involved in cellular processes such as growth, proliferation, and survival. The “memory effect” suggests that the changes induced by rapamycin, particularly in the regulation of mTOR-related processes, continue to influence the cellular behavior or function even when the drug is no longer present. As clinical trials of drugs like rapamycin are already underway and currently being tested in the context of aging, these are attractive options for targeting histone loss and genomic instability. Given that aging is a continuous process that unfolds over many years in humans, with cells having variable turnover times, tracking histone changes in specific cell types becomes imperative for a better understanding of how histones contribute to each stage of aging. The continuing advances in RNA-seq technologies, particularly single-cell omics sequencing, provide a higher resolution for dissecting epigenetic characteristics during aging and provide new leads for investigating the heterogeneity of aged cells. The creation of a spatiotemporal atlas detailing histone recruitment or eviction from the genome across multiple mammalian tissues can provide deeper insight into the epigenetic changes that are causal to aging.

## Figures and Tables

**Figure 1 cells-13-00320-f001:**
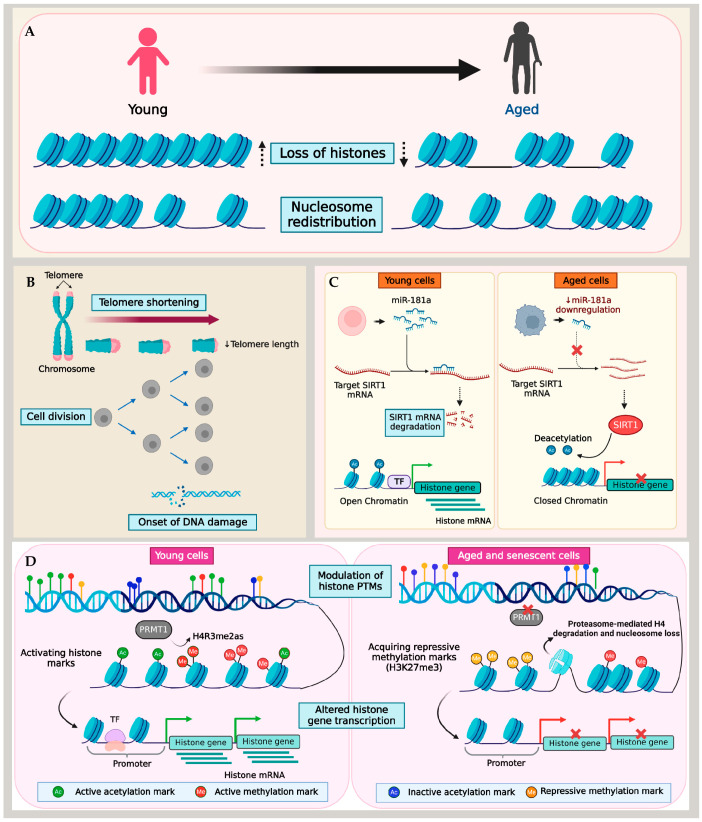
Mechanism of histone loss in aging and senescence. (**A**) Aging and senescence are characterized by the loss of histones and nucleosome redistribution. Different models show histone loss due to (**B**) telomere shortening and DNA damage, (**C**) miRNA-mediated changes to histone transcription, and (**D**) the imbalance of activating and repressive histone PTMs.

**Figure 2 cells-13-00320-f002:**
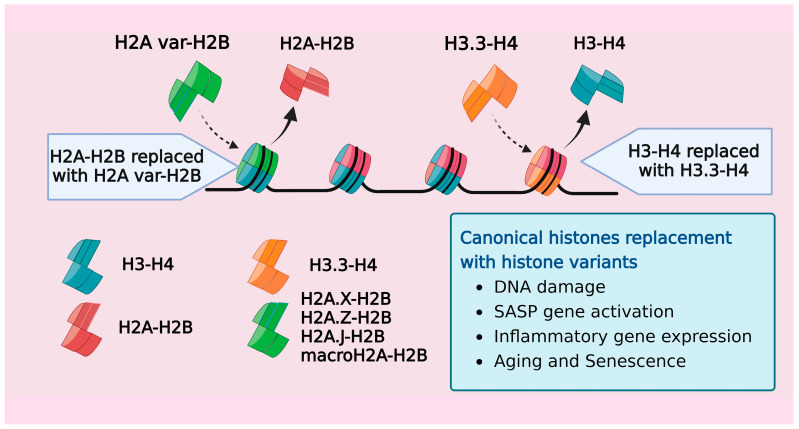
A schematic representation showing the exchange of canonical histones with variants over a gene. Histone exchange involves the replacement of both H2A and H3 for their respective histone variants in response to age-related stress stimuli.

**Figure 3 cells-13-00320-f003:**
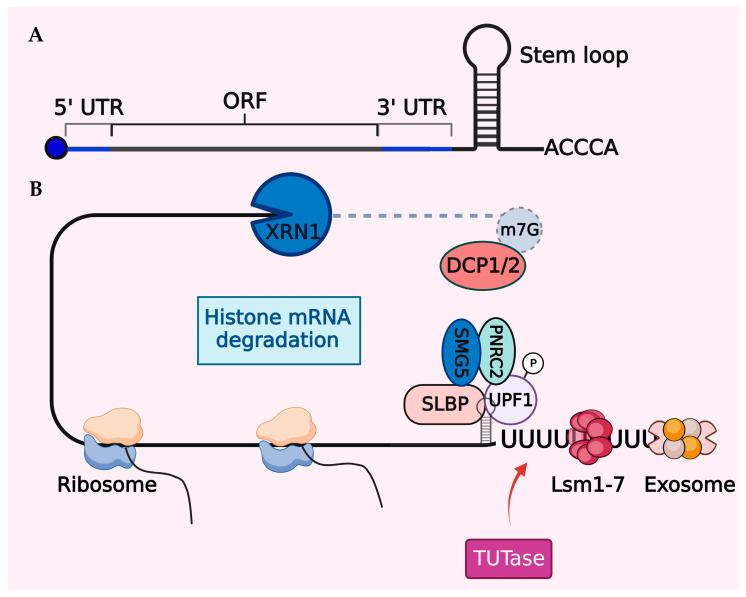
Degradation mechanism of histone mRNA. (**A**) Schematic illustration of histone mRNA showing the cap on the 5′ end, the short 5′ and 3′ UTRs, and the stem–loop at the 3′ end. (**B**) Following translation termination, phosphorylated Upf1, a crucial factor for nonsense-mediated mRNA decay (NMD), is recruited to the 3′ end of histone mRNA and subsequently, Upf1 interacts with other NMD cofactors such as SMG5 and PNRC2. Histone mRNA degradation occurs through decapping with DCP1/2 and 5′→3′ cleavage by XRN1 from the 5′ end. Additionally, TUTase-mediated oligouridylation targets histone mRNA for 3′→5′ degradation by the exosome.

**Figure 4 cells-13-00320-f004:**
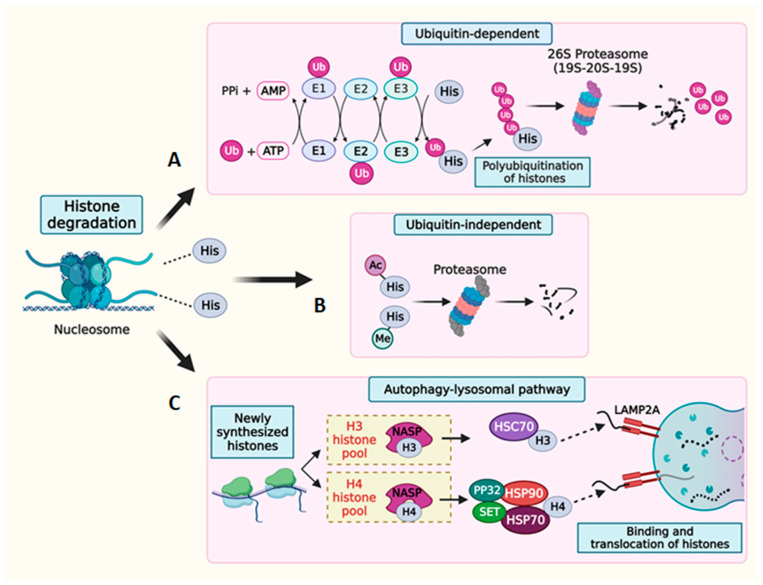
Degradation of histone proteins. Schematic representation of histone degradation via (**A**) ubiquitin-dependent or (**B**) ubiquitin-independent pathways. (**C**) Schematic representation of histone degradation mediated by the autophagy–lysosomal system. The NASP protein stabilizes a reservoir of soluble histones H3-H4 by protecting them from degradation. Chaperone-mediated autophagy involves the direct uptake of cytosolic chaperone-bound H3 and H4 into lysosomes via a translocation complex consisting of LAMP2A monomers.

**Table 1 cells-13-00320-t001:** Histone loss and changes in nucleosome occupancy in yeast aging.

Part A	Histone Reduction	Organism	Reason for Histone Alterations	Impact of Histone Loss	Reference
1	H2B	MHY103	Conditional repression of H2B mRNA synthesis using a yeast strain, with single H2B gene fused with a repressible GAL10 promoter	Cell cycle arrest, disruption of chromatin structure, disruption of nuclear segregation	Han et al., 1987 [52]
2	H4	UKY403	Conditional repression of H4 in yeast strain, with single histone H4 gene under the control of GAL1 promoter	Cell cycle arrest at G2, disruption of chromatin structure and chromosomal segregation	Kim et al., 1988 [53]
3	H4	*S. cerevisiae* UKY403 and MHY308	Conditional repression of H4 in yeast strain, with single histone H4 gene under the control of GAL1 promoter	Preferential derepression of genes at telomeric heterochromatin	Wyrick et al., 1999 [55]
4	H4	Yeast	Replicative aging	Decreased expression of histone deacetylase, Sir2, accompanied by an increase in H4K16 acetylation; transcriptional derepression at specific loci near telomeres	Dang et al., 2009 [58]
5	H3 and H2B	Yeast	Age-related histone loss	Increased transcription of histone genes with age, but depletion of histone proteins; histone occupancy is reduced by 50–75% in the aged population	Feser et al. 2010 [59]
6	H2A, H2B, H3, H4	Yeast nhp6a/b double mutant	Yeast cells with the nhp6a/b double mutation; lack HMG-box proteins Nhp6a and Nhp6b; they demonstrate many senescence-related characteristics including reduced histone content	Does not affect nucleosome spacing, but rather, changes nucleosome occupancy with the loss typically being concentrated in nucleosome-poor regions of the chromatin; global upregulation in transcription due to increased DNA accessibility	Celona et al., 2011 [60]
7	H3	YEF473A (WT) and DCB200.1	Inducing the repression of H3 conditionally in a yeast mutant strain involved deleting one H3-encoding gene, HHT1, and placing HHT2 under the regulation of the GAL1 promoter	Specific sets of nucleosomes within undergo changes in their occupancy, and nucleosomes are typically lost at gene promoters; limited histone availability results in DNA-encoded preferred nucleosome occupancy and chromatin stability	Gossett et al. 2012 [54]
8	H2A, H2B, H3, H4	Derivative strains of *S. cerevisae* BY4741/2	Yeast mutant strain lacking telomerase in senescence relocalize Rap1 transcription factor to canonical histone promoters and downregulate their expression	Rap1 binds and represses histone genes in senescent cells; Rap1 and nucleosome occupancy exhibit an inverse correlation at many genomic loci	Platt et al., 2013 [61]
9	H2A, H2B, H3, H4	Derivative strains of *S. cerevisiae* S288c (BY4741)	Replicative aging of yeast cells	Nucleosome positioning became less precise; transcriptional activation; derepression of genes that are transcriptionally silent due to loss of histones from the promoters; elevated levels of DNA strand breaks; mitochondrial DNA transfer; genomic instability is attributed to large-scale chromatin rearrangements including translocations and retrotransposition	Hu et al., 2014 [56]

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
