# Peer review of "Unraveling Histone Loss in Aging and Senescence"

_cells, 2024, doi:10.3390/cells13040320_

Round 1

Reviewer 1 Report

Comments and Suggestions for Authors

In this well written and organized review manuscript Dubey et. al. the literature concerning histone loss, differential expression, proteostasis, variant expression and genome-wide histone redistribution as a fundamental process that contributes to the processes of cell senescence and biological aging is outlined.  They begin with a description of the several decades of studying the alteration and loss of histones in non-mammalian model organisms such as yeast and C. elegans, followed by a thorough description of the later studies done in mammalian models of replicative senescence and chronological/biological aging.  The various studies are clearly listed in tabular form, making comparison straightforward.  This review is timely and is appropriate for publication in Cells, with some minor additions noted below, including a more comprehensive table of the known histones and their variants.

Line 124: it is important to note that not all studies have linked alterations of histone H1 subspecies to cell aging but rather note that it may be a reflection more of cell cycle distribution (Houde M, Shmookler Reis RJ, Goldstein S. Proportions of H1 histone subspecies in human fibroblasts shift during density-dependent growth arrest independent of replicative senescence. Exp Cell Res. 1989 Sep;184(1):256-61).

Line 218: It would be appropriate to note that in addition to alteration of nucleosomes at the serum response factor (SRF) gene whose activity is needed for mitogen-induced cell growth, that this mechanism is also linked to post-translational modification of SRF as noted in (Wheaton K, Riabowol K. Protein kinase C delta blocks immediate-early gene expression in senescent cells by inactivating serum response factor. Mol Cell Biol. 2004 Aug;24(16):7298-311).

Line 252:  The authors note the linkage between histone loss and telomere depletion.  Given that telomeres tend to cluster near the nuclear membrane that has heterochromatic character and that results in transcriptional silencing (Andrulis ED, Neiman AM, Zappulla DC, Sternglanz R. Perinuclear localization of chromatin facilitates transcriptional silencing. Nature. 1998 Aug 6;394(6693):592-5;  Galy V, Olivo-Marin JC, Scherthan H, Doye V, Rascalou N, Nehrbass U. Nuclear pore complexes in the organization of silent telomeric chromatin. Nature. 2000 Jan 6;403(6765):108-12.), perhaps the authors could include a section noting this correlation between intranuclear histone localization and histone distribution.  Such a link has also been reported during cell senescence (Raz V, Vermolen BJ, Garini Y, Onderwater JJ, Mommaas-Kienhuis MA, Koster AJ, Young IT, Tanke H, Dirks RW. The nuclear lamina promotes telomere aggregation and centromere peripheral localization during senescence of human mesenchymal stem cells. J Cell Sci. 2008 Dec 15;121(Pt 24):4018-28). Although such distribution is noted at line 351, more emphasis on localization and histone density, variant expression and modification would be informative. 

General comments:

-Section 6:  a table of canonical core and linker histones and the histone variants in humans and other mammals would be informative

-Section 7: the histone degradation section is well written and organized

Author Response

Response to Reviewer 1:

In this well written and organized review manuscript Dubey et. al. the literature concerning histone loss, differential expression, proteostasis, variant expression and genome-wide histone redistribution as a fundamental process that contributes to the processes of cell senescence and biological aging is outlined.  They begin with a description of the several decades of studying the alteration and loss of histones in non-mammalian model organisms such as yeast and C. elegans, followed by a thorough description of the later studies done in mammalian models of replicative senescence and chronological/biological aging.  The various studies are clearly listed in tabular form, making comparison straightforward.  This review is timely and is appropriate for publication in Cells, with some minor additions noted below, including a more comprehensive table of the known histones and their variants.

We thank the reviewer for dedicating time and effort to review our manuscript. The constructive feedback provided has been immensely valuable in enhancing the content and presentation of our review. Our responses, along with necessary revisions, are incorporated as track changes and referenced accordingly.  

Comment 1: Line 124: it is important to note that not all studies have linked alterations of histone H1 subspecies to cell aging but rather note that it may be a reflection more of cell cycle distribution (Houde M, Shmookler Reis RJ, Goldstein S. Proportions of H1 histone subspecies in human fibroblasts shift during density-dependent growth arrest independent of replicative senescence. Exp Cell Res. 1989 Sep;184(1):256-61).

Response 1: We are grateful to the reviewer for drawing our attention to this paper by Houde et al., which we think makes an excellent contribution toward understanding the changes in H1 variants as a function of time of density-dependent growth rather than replicative age. We agree that this is a key point that is relevant to the current review and have included this noteworthy information in the review.

We have added the following statement in the revised manuscript:

“Houde and colleagues [1] provide a novel perspective on H1 expression changes in cultured human fibroblasts that exhibit a progressive cell cycle elongation. Their findings revealed that cells in early (28-35 mean population doubling; MPD) and late (65-70 MPD) passages, maintained in a confluent state at 0 and 6 weeks, exhibited a similar shift in gene expression of H1 variants. Notably, this involved a reduction in H1B and a concurrent increase in H10 and H1A. These data suggest that changes in the expression levels of histone H1 variants occurred as a function of time of density-dependent growth rather than replicative age.” (Page 6, Lines 182-189)

Comment 2: Line 218: It would be appropriate to note that in addition to alteration of nucleosomes at the serum response factor (SRF) gene whose activity is needed for mitogen-induced cell growth, that this mechanism is also linked to post-translational modification of SRF as noted in (Wheaton K, Riabowol K. Protein kinase C delta blocks immediate-early gene expression in senescent cells by inactivating serum response factor. Mol Cell Biol. 2004 Aug;24(16):7298-311).

Response 2: We thank the reviewer for bringing this paper by Wheaton and Riabowol to our attention. This study underscores the significance of protein kinase C-δ induced phosphorylation, leading to the inactivation of SRF, a key transcriptional activator of immediate-early gene promoters in senescent cells. In our manuscript, we highlight that the increased nucleosome occupancy as a factor leading to SRF depletion in the aging liver, resulting in derepression of its target genes. However, as the reviewer noted, this might not be the sole determinant for the downregulation of SRF gene and the consequent derepression of its targets. Additional factors, such as post-translational phosphorylation of SRF, could also compromise its DNA binding activity. We have included this information in our revised manuscript.

“Although this study focused on the decline in SRF activity in the aging liver due to altered nucleosome occupancy, it is important to note that other mechanisms have been reported for repressed SRF activity, including nuclear exclusion, protein kinase C-δ induced phosphorylation, and inactivation of SRF in senescent cells [2,3].” (Page 8, Lines 297-301)

Comment 3: Line 252:  The authors note the linkage between histone loss and telomere depletion.  Given that telomeres tend to cluster near the nuclear membrane that has heterochromatic character and that results in transcriptional silencing (Andrulis ED, Neiman AM, Zappulla DC, Sternglanz R. Perinuclear localization of chromatin facilitates transcriptional silencing. Nature. 1998 Aug 6;394(6693):592-5;  Galy V, Olivo-Marin JC, Scherthan H, Doye V, Rascalou N, Nehrbass U. Nuclear pore complexes in the organization of silent telomeric chromatin. Nature. 2000 Jan 6;403(6765):108-12.), perhaps the authors could include a section noting this correlation between intranuclear histone localization and histone distribution.  Such a link has also been reported during cell senescence (Raz V, Vermolen BJ, Garini Y, Onderwater JJ, Mommaas-Kienhuis MA, Koster AJ, Young IT, Tanke H, Dirks RW. The nuclear lamina promotes telomere aggregation and centromere peripheral localization during senescence of human mesenchymal stem cells. J Cell Sci. 2008 Dec 15;121(Pt 24):4018-28). Although such distribution is noted at line 351, more emphasis on localization and histone density, variant expression and modification would be informative. 

Response 3: We thank the reviewer for suggesting these references. We agree that age-related telomere modulations are relevant to this review, and we have now added these three paragraphs with references in Section 5:

“Telomere shortening and dysfunction are hallmarks of cellular aging and senescence. Despite their crucial role in maintaining chromosomal stability, the regulatory mechanisms governing telomeres during cellular aging remain poorly understood. Situated at the termini of linear chromosomes, telomeres form complexes of TTAGGG nucleotide repeats and proteins that regulate their functions, shielding them from recognition by the cell’s repair mechanism as double stranded DNA breaks (DSBs). These proteins, including the telomerase (TERT) enzyme, histones, and the Shelterin complex, are critical for regulating telomere length and preventing chromosomal end fusion [4,5].

Telomeres typically exhibit a heterochromatin structure, and a widespread phenomenon known as telomere position effect results in low expression levels or transcriptional silencing of genes within or near telomeres [6]. Additionally, telomeres tend to spatially organize at the nuclear periphery, a zone of transcriptional repression, in a cell-cycle-dependent manner [7-9] and therefore experience transcriptional repression of genes. However, as cells approached senescence a spatial overlap of lamina intranuclear structures with telomere was observed [10]. During senescence, as the nuclear lamina's organization gets disrupted, telomeres tend to form large aggregates lacking TERT. These telomere aggregates accumulated histone γ-H2AX, a classical marker of DSBs and telomere shortening, in senescent cells [10].

Furthermore, the formation of senescence-associated heterochromatin foci (SAHF), representing facultative heterochromatin domains, correlates with telomere shortening in cells entering senescence [11,12]. SAHF contain domains with di-or tri-methylated lysine 9 of histone H3 (H3K9me2/3), a histone H2A variant (macroH2A), and heterochromatin protein 1 (HP1) proteins [11,13,14]. Additionally, epigenetic modifications such as histone methylation in the telomere region and TERT demethylation in humans play significant roles in maintaining heterochromatin, transcriptional silencing at telomeres, and telomerase inactivation. Preserving the telomere structure and ensuring transcriptional silencing are critical to preventing premature aging [15,16].” (Pages 12, Lines 354-380)

General comments:

Comment 4: -Section 6:  a table of canonical core and linker histones and the histone variants in humans and other mammals would be informative

Response 4: We agree with the reviewer that a comprehensive table for all histones and their variants should be included in the current review. A table listing all the histones and histone variants from humans has been added on Pages 18-22 Table 3.

Comment 5: -Section 7: the histone degradation section is well written and organized

Response 5: Thanks. We appreciate your positive feedback.

  1. Houde, M.; Shmookler Reis, R.J.; Goldstein, S. Proportions of H1 histone subspecies in human fibroblasts shift during density-dependent growth arrest independent of replicative senescence. Exp Cell Res 1989, 184, 256-261, doi:10.1016/0014-4827(89)90384-4.
  2. Wheaton, K.; Riabowol, K. Protein kinase C delta blocks immediate-early gene expression in senescent cells by inactivating serum response factor. Mol Cell Biol 2004, 24, 7298-7311, doi:10.1128/MCB.24.16.7298-7311.2004.
  3. Ding, W.; Gao, S.; Scott, R.E. Senescence represses the nuclear localization of the serum response factor and differentiation regulates its nuclear localization with lineage specificity. J Cell Sci 2001, 114, 1011-1018, doi:10.1242/jcs.114.5.1011.
  4. de Lange, T. Shelterin: the protein complex that shapes and safeguards human telomeres. Genes Dev 2005, 19, 2100-2110, doi:10.1101/gad.1346005.
  5. Bar, C.; Blasco, M.A. Telomeres and telomerase as therapeutic targets to prevent and treat age-related diseases. F1000Res 2016, 5, doi:10.12688/f1000research.7020.1.
  6. Gottschling, D.E.; Aparicio, O.M.; Billington, B.L.; Zakian, V.A. Position effect at S. cerevisiae telomeres: reversible repression of Pol II transcription. Cell 1990, 63, 751-762, doi:10.1016/0092-8674(90)90141-z.
  7. Weierich, C.; Brero, A.; Stein, S.; von Hase, J.; Cremer, C.; Cremer, T.; Solovei, I. Three-dimensional arrangements of centromeres and telomeres in nuclei of human and murine lymphocytes. Chromosome Res 2003, 11, 485-502, doi:10.1023/a:1025016828544.
  8. Andrulis, E.D.; Neiman, A.M.; Zappulla, D.C.; Sternglanz, R. Perinuclear localization of chromatin facilitates transcriptional silencing. Nature 1998, 394, 592-595, doi:10.1038/29100.
  9. Galy, V.; Olivo-Marin, J.C.; Scherthan, H.; Doye, V.; Rascalou, N.; Nehrbass, U. Nuclear pore complexes in the organization of silent telomeric chromatin. Nature 2000, 403, 108-112, doi:10.1038/47528.
  10. Raz, V.; Vermolen, B.J.; Garini, Y.; Onderwater, J.J.; Mommaas-Kienhuis, M.A.; Koster, A.J.; Young, I.T.; Tanke, H.; Dirks, R.W. The nuclear lamina promotes telomere aggregation and centromere peripheral localization during senescence of human mesenchymal stem cells. J Cell Sci 2008, 121, 4018-4028, doi:10.1242/jcs.034876.
  11. Narita, M.; Nunez, S.; Heard, E.; Narita, M.; Lin, A.W.; Hearn, S.A.; Spector, D.L.; Hannon, G.J.; Lowe, S.W. Rb-mediated heterochromatin formation and silencing of E2F target genes during cellular senescence. Cell 2003, 113, 703-716, doi:10.1016/s0092-8674(03)00401-x.
  12. Narita, M.; Narita, M.; Krizhanovsky, V.; Nunez, S.; Chicas, A.; Hearn, S.A.; Myers, M.P.; Lowe, S.W. A novel role for high-mobility group a proteins in cellular senescence and heterochromatin formation. Cell 2006, 126, 503-514, doi:10.1016/j.cell.2006.05.052.
  13. Zhang, R.; Poustovoitov, M.V.; Ye, X.; Santos, H.A.; Chen, W.; Daganzo, S.M.; Erzberger, J.P.; Serebriiskii, I.G.; Canutescu, A.A.; Dunbrack, R.L.; et al. Formation of MacroH2A-containing senescence-associated heterochromatin foci and senescence driven by ASF1a and HIRA. Dev Cell 2005, 8, 19-30, doi:10.1016/j.devcel.2004.10.019.
  14. Kreiling, J.A.; Tamamori-Adachi, M.; Sexton, A.N.; Jeyapalan, J.C.; Munoz-Najar, U.; Peterson, A.L.; Manivannan, J.; Rogers, E.S.; Pchelintsev, N.A.; Adams, P.D.; et al. Age-associated increase in heterochromatic marks in murine and primate tissues. Aging Cell 2011, 10, 292-304, doi:10.1111/j.1474-9726.2010.00666.x.
  15. Smeal, T.; Claus, J.; Kennedy, B.; Cole, F.; Guarente, L. Loss of transcriptional silencing causes sterility in old mother cells of S. cerevisiae. Cell 1996, 84, 633-642, doi:10.1016/s0092-8674(00)81038-7.
  16. Kozak, M.L.; Chavez, A.; Dang, W.; Berger, S.L.; Ashok, A.; Guo, X.; Johnson, F.B. Inactivation of the Sas2 histone acetyltransferase delays senescence driven by telomere dysfunction. EMBO J 2010, 29, 158-170, doi:10.1038/emboj.2009.314.

Reviewer 2 Report

Comments and Suggestions for Authors

This is an interesting review, however, it has to be improved before acceptance.

1)      The authors should prepare a good graphical abstract (GA) at the end of the introduction. They can use Figure 1 (move it to the end of the introduction) along with modification.  

2)      It is known during aging, the PTM pattern is changed. The most important question is how (molecular mechanism) these histone losses (or changes in histone modification) lead to aging not cancer. The authors need to discuss this part comprehensively.

3)      Although the title of the manuscript is The Study of Histone Loss in Aging, it seems the influence of telomere shortening on aging is higher than the others. It would be nice if the authors discussed it in a section.

4)      What is the unstable H4 (figure 1D- right)? Histone proteins are considered stable proteins. It may be referred to as histone octamer due to electrostatic repulsion or ion unbalancing.

5)      For active acetylation mark, the authors could give an example H4K16AC (they can use this paper as ref.  https://academic.oup.com/nar/article/39/5/1680/2409413?login=true    )  

6)      The replacement of histones by variant types could happen during embryogenesis and differentiation. E.g. H2A to H2A.Z which is required for mammalian development. How do the authors distinguish it from gaining?

Author Response

Response to Reviewer 2

This is an interesting review, however, it has to be improved before acceptance.

We thank the reviewer for the helpful review that will significantly improve the manuscript. Here, we provide a detailed point-by-point response to the reviewer’s comments along with the revised manuscript.

Comment 1: The authors should prepare a good graphical abstract (GA) at the end of the introduction. They can use Figure 1 (move it to the end of the introduction) along with modification. 

Response 1: We thank the reviewer for the suggestion. A graphical abstract has been included in the revised manuscript.

Comment 2: It is known during aging, the PTM pattern is changed. The most important question is how (molecular mechanism) these histone losses (or changes in histone modification) lead to aging not cancer. The authors need to discuss this part comprehensively.

Response 2: We appreciate the reviewer’s comment. In the current review, we explore the dynamics of histone loss and the replacement of canonical histones by variants in the context of aging and senescence. In response to the reviewer's suggestion, we have integrated the following lines to delineate the distinction between histone loss or substitution by variants, contributing to either aging or cancer.

“While the acquisition of senescence has been viewed as an alternative pathway to prevent cancer, the prolonged accumulation of senescent cells can paradoxically promote cancer development [1,2]. The emergence of SASP factors during senescence can impact the surrounding cells via alterations of cellular microenvironments, establishing chronic inflammation and fostering conditions conducive to cancer [3]. Interestingly, histone depletion associated with aging and senescence has been inversely linked to malignancy, suggesting that histone loss can facilitate cell proliferation arrest, ultimately contributing to tumor suppression [4]. Unlike histone loss, histone mutations, known as oncohistones, are implicated in promoting cancers [5]. However, given that many age-dependent changes in the cellular epigenome resemble those observed in cancer, the epigenetic reprogramming occurring during aging may predispose individuals to cancer development [6]. Therefore, gaining a deeper understanding of age-related epigenomic changes holds the potential to elucidate the underlying causes of cancer." (Page 7; Lines 231-243)

"Regulated deposition of histone variants plays a crucial role in preserving the senescent state of the cell, acting as a barrier against the development of cancer [7,8]. Mutations in these histone variants can result in severe defects and are commonly observed in association with various cancers [8]. The intricate interplay between these histone variants in senescent cells underscores their significance in maintaining genomic integrity, preventing uncontrolled cell proliferation, and the potential transformation into cancerous cells.” (Page 16, Lines 502-508)

Comment 3: Although the title of the manuscript is The Study of Histone Loss in Aging, it seems the influence of telomere shortening on aging is higher than the others. It would be nice if the authors discussed it in a section.

Response 3: We recognize the pivotal role of telomeres in the aging process, and we have incorporated three paragraphs that describe the age-related modulation of telomeres, along with the accompanying alterations in the histone composition of the telomeric region. Additional paragraphs detailing the importance of telomeric aging are included with the following statements in Section 5 (Pages 11-12, Lines 336-362)

“Telomere shortening and dysfunction are hallmarks of cellular aging and senescence. Despite their crucial role in maintaining chromosomal stability, the regulatory mechanisms governing telomeres during cellular aging remain poorly understood. Situated at the termini of linear chromosomes, telomeres form complexes of TTAGGG nucleotide repeats and proteins that regulate their functions, shielding them from recognition by the cell’s repair mechanism as double stranded DNA breaks (DSBs). These proteins, including the telomerase (TERT) enzyme, histones, and the Shelterin complex, are critical for regulating telomere length and preventing chromosomal end fusion [9,10].

Telomeres typically exhibit a heterochromatin structure, and a widespread phenomenon known as telomere position effect results in low expression levels or transcriptional silencing of genes within or near telomeres [11]. Additionally, telomeres tend to spatially organize at the nuclear periphery, a zone of transcriptional repression, in a cell-cycle-dependent manner [12-14] and therefore experience transcriptional repression of genes. However, as cells approached senescence a spatial overlap of lamina intranuclear structures with telomere was observed [15]. During senescence, as the nuclear lamina's organization gets disrupted, telomeres tend to form large aggregates lacking TERT. These telomere aggregates accumulated histone γ-H2AX, a classical marker of DSBs and telomere shortening, in senescent cells [15].

Furthermore, the formation of senescence-associated heterochromatin foci (SAHF), representing facultative heterochromatin domains, correlates with telomere shortening in cells entering senescence [16,17]. SAHF contain domains with di-or tri-methylated lysine 9 of histone H3 (H3K9me2/3), a histone H2A variant (macroH2A), and heterochromatin protein 1 (HP1) proteins [16,18,19]. Additionally, epigenetic modifications such as histone methylation in the telomere region and TERT demethylation in humans play significant roles in maintaining heterochromatin, transcriptional silencing at telomeres, and telomerase inactivation. Preserving the telomere structure and ensuring transcriptional silencing are critical to preventing premature aging [20,21].” (Pages 11-12, Lines 336-362)

Comment 4: What is the unstable H4 (figure 1D- right)? Histone proteins are considered stable proteins. It may be referred to as histone octamer due to electrostatic repulsion or ion unbalancing.

Response 4: We acknowledge and appreciate the reviewer’s observation, and we apologize for the mislabeling of Figure 1D in our manuscript. To clarify, the left panel of Figure 1D accurately represents PRMT1-mediated dimethylation of histone H4 at arginine 3 (H4R3me2as), contributing to the maintenance of histone H4 stability. Conversely, the right panel illustrates the consequences of PRMT1 deficiency, depicting the loss of H4 stability, leading to proteasome-mediated H4 degradation and nucleosome loss [22]. We have re-edited the figure 1D, updating the labeling to now read as 'proteasome-mediated H4 degradation and nucleosome loss’, providing a more accurate representation of the depicted processes.

Comment 5: For active acetylation mark, the authors could give an example H4K16AC (they can use this paper as ref: https://academic.oup.com/nar/article/39/5/1680/2409413

Response 5: We thank the reviewer for the reference and it has been included in the revised manuscript. (Page 2, Line 93)

Comment 6: The replacement of histones by variant types could happen during embryogenesis and differentiation. E.g. H2A to H2A.Z which is required for mammalian development. How do the authors distinguish it from gaining?

Response 6: We thank the reviewer for their question regarding the significance of histone replacement by variant forms across different life stages in organisms. In mammals, core histone proteins have many sequence variants, characterized by either minor sequence differences (e.g., the canonical H3.1 and H3.2, and the variant H3.3) or substantial structural differences (e.g., macroH2A, the centromere-specific protein CENP-A). These variants exhibit replication-independent, tissue-specific expression levels with varying temporal and developmental expression profiles.

In response to the reviewer's inquiry, considering the transition from embryogenesis and differentiation to the onset of aging, the organisms undergo drastic changes in their epigenetic landscape. For instance, histone variant H2A.Z plays diverse and contrasting roles and are crucial for embryogenesis and differentiation in various organisms such as mice, metazoans, and C. elegans [23-25]. However, various studies on aging and histone variants have consistently revealed the enrichment of H2A.Z in the aging genome [26]. A temporal evaluation of H2A.Z showed varied expression levels at different developmental stages [24,27]. H2A.Z has been linked to transcriptional activation, repression, cell cycle control, DNA replication, and DNA damage at various life stages of the organism [28]. Thus, the replacement of histones by variant forms is not only integral to the extensive epigenetic reprogramming during the initial stages of germ cell lineage, embryogenesis, and cell differentiation but also actively participates in the epigenetic changes associated with aging.

  1. Coppe, J.P.; Desprez, P.Y.; Krtolica, A.; Campisi, J. The senescence-associated secretory phenotype: the dark side of tumor suppression. Annu Rev Pathol 2010, 5, 99-118, doi:10.1146/annurev-pathol-121808-102144.
  2. Wang, L.; Lankhorst, L.; Bernards, R. Exploiting senescence for the treatment of cancer. Nat Rev Cancer 2022, 22, 340-355, doi:10.1038/s41568-022-00450-9.
  3. Domen, A.; Deben, C.; Verswyvel, J.; Flieswasser, T.; Prenen, H.; Peeters, M.; Lardon, F.; Wouters, A. Cellular senescence in cancer: clinical detection and prognostic implications. J Exp Clin Cancer Res 2022, 41, 360, doi:10.1186/s13046-022-02555-3.
  4. Ivanov, A.; Pawlikowski, J.; Manoharan, I.; van Tuyn, J.; Nelson, D.M.; Rai, T.S.; Shah, P.P.; Hewitt, G.; Korolchuk, V.I.; Passos, J.F.; et al. Lysosome-mediated processing of chromatin in senescence. J Cell Biol 2013, 202, 129-143, doi:10.1083/jcb.201212110.
  5. Amatori, S.; Tavolaro, S.; Gambardella, S.; Fanelli, M. The dark side of histones: genomic organization and role of oncohistones in cancer. Clin Epigenetics 2021, 13, 71, doi:10.1186/s13148-021-01057-x.
  6. Gautrey, H.E.; van Otterdijk, S.D.; Cordell, H.J.; Newcastle 85+ Study Core, T.; Mathers, J.C.; Strathdee, G. DNA methylation abnormalities at gene promoters are extensive and variable in the elderly and phenocopy cancer cells. FASEB J 2014, 28, 3261-3272, doi:10.1096/fj.13-246173.
  7. Duarte, L.F.; Young, A.R.; Wang, Z.; Wu, H.A.; Panda, T.; Kou, Y.; Kapoor, A.; Hasson, D.; Mills, N.R.; Ma'ayan, A.; et al. Histone H3.3 and its proteolytically processed form drive a cellular senescence programme. Nat Commun 2014, 5, 5210, doi:10.1038/ncomms6210.
  8. Rai, T.S.; Cole, J.J.; Nelson, D.M.; Dikovskaya, D.; Faller, W.J.; Vizioli, M.G.; Hewitt, R.N.; Anannya, O.; McBryan, T.; Manoharan, I.; et al. HIRA orchestrates a dynamic chromatin landscape in senescence and is required for suppression of neoplasia. Genes Dev 2014, 28, 2712-2725, doi:10.1101/gad.247528.114.
  9. de Lange, T. Shelterin: the protein complex that shapes and safeguards human telomeres. Genes Dev 2005, 19, 2100-2110, doi:10.1101/gad.1346005.
  10. Bar, C.; Blasco, M.A. Telomeres and telomerase as therapeutic targets to prevent and treat age-related diseases. F1000Res 2016, 5, doi:10.12688/f1000research.7020.1.
  11. Gottschling, D.E.; Aparicio, O.M.; Billington, B.L.; Zakian, V.A. Position effect at S. cerevisiae telomeres: reversible repression of Pol II transcription. Cell 1990, 63, 751-762, doi:10.1016/0092-8674(90)90141-z.
  12. Weierich, C.; Brero, A.; Stein, S.; von Hase, J.; Cremer, C.; Cremer, T.; Solovei, I. Three-dimensional arrangements of centromeres and telomeres in nuclei of human and murine lymphocytes. Chromosome Res 2003, 11, 485-502, doi:10.1023/a:1025016828544.
  13. Andrulis, E.D.; Neiman, A.M.; Zappulla, D.C.; Sternglanz, R. Perinuclear localization of chromatin facilitates transcriptional silencing. Nature 1998, 394, 592-595, doi:10.1038/29100.
  14. Galy, V.; Olivo-Marin, J.C.; Scherthan, H.; Doye, V.; Rascalou, N.; Nehrbass, U. Nuclear pore complexes in the organization of silent telomeric chromatin. Nature 2000, 403, 108-112, doi:10.1038/47528.
  15. Raz, V.; Vermolen, B.J.; Garini, Y.; Onderwater, J.J.; Mommaas-Kienhuis, M.A.; Koster, A.J.; Young, I.T.; Tanke, H.; Dirks, R.W. The nuclear lamina promotes telomere aggregation and centromere peripheral localization during senescence of human mesenchymal stem cells. J Cell Sci 2008, 121, 4018-4028, doi:10.1242/jcs.034876.
  16. Narita, M.; Nunez, S.; Heard, E.; Narita, M.; Lin, A.W.; Hearn, S.A.; Spector, D.L.; Hannon, G.J.; Lowe, S.W. Rb-mediated heterochromatin formation and silencing of E2F target genes during cellular senescence. Cell 2003, 113, 703-716, doi:10.1016/s0092-8674(03)00401-x.
  17. Narita, M.; Narita, M.; Krizhanovsky, V.; Nunez, S.; Chicas, A.; Hearn, S.A.; Myers, M.P.; Lowe, S.W. A novel role for high-mobility group a proteins in cellular senescence and heterochromatin formation. Cell 2006, 126, 503-514, doi:10.1016/j.cell.2006.05.052.
  18. Zhang, R.; Poustovoitov, M.V.; Ye, X.; Santos, H.A.; Chen, W.; Daganzo, S.M.; Erzberger, J.P.; Serebriiskii, I.G.; Canutescu, A.A.; Dunbrack, R.L.; et al. Formation of MacroH2A-containing senescence-associated heterochromatin foci and senescence driven by ASF1a and HIRA. Dev Cell 2005, 8, 19-30, doi:10.1016/j.devcel.2004.10.019.
  19. Kreiling, J.A.; Tamamori-Adachi, M.; Sexton, A.N.; Jeyapalan, J.C.; Munoz-Najar, U.; Peterson, A.L.; Manivannan, J.; Rogers, E.S.; Pchelintsev, N.A.; Adams, P.D.; et al. Age-associated increase in heterochromatic marks in murine and primate tissues. Aging Cell 2011, 10, 292-304, doi:10.1111/j.1474-9726.2010.00666.x.
  20. Smeal, T.; Claus, J.; Kennedy, B.; Cole, F.; Guarente, L. Loss of transcriptional silencing causes sterility in old mother cells of S. cerevisiae. Cell 1996, 84, 633-642, doi:10.1016/s0092-8674(00)81038-7.
  21. Kozak, M.L.; Chavez, A.; Dang, W.; Berger, S.L.; Ashok, A.; Guo, X.; Johnson, F.B. Inactivation of the Sas2 histone acetyltransferase delays senescence driven by telomere dysfunction. EMBO J 2010, 29, 158-170, doi:10.1038/emboj.2009.314.
  22. Lin, C.; Li, H.; Liu, J.; Hu, Q.; Zhang, S.; Zhang, N.; Liu, L.; Dai, Y.; Cao, D.; Li, X.; et al. Arginine hypomethylation-mediated proteasomal degradation of histone H4-an early biomarker of cellular senescence. Cell Death Differ 2020, 27, 2697-2709, doi:10.1038/s41418-020-0562-8.
  23. Liu, X.; Zhang, J.; Zhou, J.; Bu, G.; Zhu, W.; He, H.; Sun, Q.; Yu, Z.; Xiong, W.; Wang, L.; et al. Hierarchical Accumulation of Histone Variant H2A.Z Regulates Transcriptional States and Histone Modifications in Early Mammalian Embryos. Adv Sci (Weinh) 2022, 9, e2200057, doi:10.1002/advs.202200057.
  24. Dijkwel, Y.; Tremethick, D.J. The Role of the Histone Variant H2A.Z in Metazoan Development. J Dev Biol 2022, 10, doi:10.3390/jdb10030028.
  25. Whittle, C.M.; McClinic, K.N.; Ercan, S.; Zhang, X.; Green, R.D.; Kelly, W.G.; Lieb, J.D. The genomic distribution and function of histone variant HTZ-1 during C. elegans embryogenesis. PLoS Genet 2008, 4, e1000187, doi:10.1371/journal.pgen.1000187.
  26. Stefanelli, G.; Azam, A.B.; Walters, B.J.; Brimble, M.A.; Gettens, C.P.; Bouchard-Cannon, P.; Cheng, H.M.; Davidoff, A.M.; Narkaj, K.; Day, J.J.; et al. Learning and Age-Related Changes in Genome-wide H2A.Z Binding in the Mouse Hippocampus. Cell Rep 2018, 22, 1124-1131, doi:10.1016/j.celrep.2018.01.020.
  27. Raja, D.A.; Subramaniam, Y.; Aggarwal, A.; Gotherwal, V.; Babu, A.; Tanwar, J.; Motiani, R.K.; Sivasubbu, S.; Gokhale, R.S.; Natarajan, V.T. Histone variant dictates fate biasing of neural crest cells to melanocyte lineage. Development 2020, 147, doi:10.1242/dev.182576.
  28. Colino-Sanguino, Y.; Clark, S.J.; Valdes-Mora, F. The H2A.Z-nucleosome code in mammals: emerging functions. Trends Genet 2022, 38, 516, doi:10.1016/j.tig.2022.02.004.

Reviewer 3 Report

Comments and Suggestions for Authors

Please find below my comments and suggestions.

Line 55. “nucleosome core comprises 146 base pairs of genomic DNA”

147 base pairs perhaps?

Line 224. “localized alterations in H3 occupancy were observed in different tissues and cells during aging” - please provide more details (which loci in which tissues, etc)

Table 2, line 2, column “Organism or cell line” - please replace “human” with “human fibroblast”.

What about aging-associated histone loss in mammalian cells other than fibroblasts, muscle stem cells, T-cells, RPE cells, HUVECs, and epidermal melanocytes discussed in this review? Do the authors leave this info beyond the scope of the current review or yet nothing is known on this subject?

I wonder how forced overexpression of histones (including overexpression only of particular variants) in cells in vitro affects markers of ageing, apart from the works in yeasts discussed in this review. Is anything known?

Line 617. “the restoration of histone expression via treatment with pharmacological compounds, including rapamycin, metformin, and resveratrol”. Is anything known about other geroprotectors?

Line 674. “due to the “memory effect” of chronic rapamycin” - could you please explain. What does it mean?

What about sirtuins and histones in ageing? Perhaps it worth briefly discussing, giving the role of sirtuins in ageing.

Is anything known about mutations affecting histone degradation and PTM, and their role in ageing phenotype?

Overall, I find this review comprehensive and brilliantly written.

Author Response

Response to Reviewer 3

Please find below my comments and suggestions.

We sincerely thank the reviewer for their positive feedback about our manuscript. We have provided a comprehensive response to each point raised, and the revised manuscript reflects the changes incorporated and author edits.

Comment 1: Line 55. “nucleosome core comprises 146 base pairs of genomic DNA”

147 base pairs perhaps?

Response 1: We apologize for the oversight. The change has been made in the revised manuscript. (Page 2; Line 91)

Comment 2: Line 224. “localized alterations in H3 occupancy were observed in different tissues and cells during aging” - please provide more details (which loci in which tissues, etc)

Response 2: This sentence was modified from “localized alterations in H3 occupancy were observed in different tissues and cells during aging” to “localized alterations in H3 occupancy were observed in all the four tissue types and cultured neural stem cells during aging.” (Page 9, Lines 307-308)

We agree with the reviewer that this point needs to be elaborated further. So, we included the following information to describe the specific areas undergoing age-related changes in H3 occupancy with the following statement:

 “Notably, the distal regions located 5-500 kb away from annotated transcriptional start sites exhibited significant remodeling of H3 occupancy, both upstream and downstream. The sites proximal to genes, particularly within intronic regions, consistently demonstrated robust nucleosome enrichment. Within aging chromatin, the distinct changes observed in distally located nucleosomes suggest a differential occupancy of nucleosomes in regulatory elements, particularly the Forkhead transcription factors which are critical regulators of DNA remodeling. Additionally, the repositioning of nucleosomes triggered transcriptional changes in inflammatory transcription factors, such as STAT6 and IRF8.” (Page 9, Lines 309-317)

Comment 3: Table 2, line 2, column “Organism or cell line” - please replace “human” with “human fibroblast”.

Response 3: Thanks for the suggestion. The change has been made from “human” to “human fibroblast”.

Comment 4: What about aging-associated histone loss in mammalian cells other than fibroblasts, muscle stem cells, T-cells, RPE cells, HUVECs, and epidermal melanocytes discussed in this review? Do the authors leave this info beyond the scope of the current review or yet nothing is known on this subject?

Response 4: The reviewer raises an important question about how widespread is the phenomenon of histone loss in mammalian cells. We have addressed this to the best of our abilities by comprehensively reviewing the literature, spanning from early investigations into age-related histone loss and changes in nucleosome occupancy in mammalian models to the most recent publications (Table 2). However, this phenomenon is not as widely researched in mammals as it is in yeast, especially concerning the mechanisms driving histone loss and relocation, and we mention this on Page 11, Lines 333-339. There remains a gap in our understanding of histone changes in mammalian models during aging. This review aims to address and comprehensively present the existing research in this field.

Comment 5: I wonder how forced overexpression of histones (including overexpression only of particular variants) in cells in vitro affects markers of ageing, apart from the works in yeasts discussed in this review. Is anything known?

Response 5: We thank the reviewer for this question. In mammals, studying the effects of histone overexpression is largely hampered by having to deal with the large number of genes that encode for both canonical and variant histones. Studies in mammalian and fly models seen in the current review have mainly adopted two strategies for histone restoration, either via histone regulatory molecules or by the use of pharmacological drugs. These studies showed that the restoration of histones impacted the cells positively, restored cell function, reduced replicative stress, and provided geroprotection. Despite the positive outcomes observed in these studies, there is a notable absence of research evaluating the impact of overexpression of histones or histone variants on various aging markers.

Comment 6: Line 617. “the restoration of histone expression via treatment with pharmacological compounds, including rapamycin, metformin, and resveratrol”. Is anything known about other geroprotectors?

Response 6: The use of geroprotectors to restore histones is a recent finding, and although, a promising avenue to reverse age-related histone loss, there are several knowledge gaps in the field. This part is included in the conclusion and future perspectives section.

Comment 7: Line 674. “due to the “memory effect” of chronic rapamycin” - could you please explain. What does it mean?

Response 7: The term "memory effect" in the context of chronic rapamycin refers to a phenomenon where the cellular or physiological effects of rapamycin persist even after the drug has been discontinued. In other words, the drug has a lasting impact on the system that persists beyond the period of active treatment. This information is included in the revised manuscript to better describe “memory effect” with the following statement (Page 29; Lines 801-806):

“Rapamycin is known for its ability to inhibit the mechanistic target of rapamycin (mTOR) pathway, which is involved in cellular processes such as growth, proliferation, and survival. The "memory effect" suggests that the changes induced by rapamycin, particularly in the regulation of mTOR-related processes, continue to influence cellular behavior or function even when the drug is no longer present.” (Page 29; Lines 801-806)

Comment 8: What about sirtuins and histones in ageing? Perhaps it worth briefly discussing, giving the role of sirtuins in ageing.

Response 8: We appreciate the reviewer's suggestion. However, the role of sirtuins in aging has been extensively reviewed [1,2]. Among the various sirtuins, SIRT1 specifically stands out for its direct regulation of histone gene expression, facilitated by its recruitment to histone gene promoters through an NPAT-dependent mechanism. The study by Kim et al., [3] addressing this aspect, has been incorporated into our manuscript. While sirtuins indeed play a crucial role in regulating various facets of chromatin biology, encompassing transcription, recombination, and genome stability through histone modifications and epigenetic reprogramming, we feel the role of sirtuins in aging is outside the scope of the current review.

Comment 9: Is anything known about mutations affecting histone degradation and PTM, and their role in ageing phenotype?

Response 9: We thank the reviewer for this question. Cellular histone levels are tightly controlled via regulation of their synthesis as well as their degradation, especially during cellular processes like DNA synthesis and proliferation. Several studies have identified amino acid residues in histone proteins that undergo PTMs that are critical for the efficient degradation of histones [4,5]. Mutations in these sites impact the degradation of histones, but we did not find any specific studies delving into the consequences of mutations in these key residues and their implications for histone degradation.

For the scope of our review, we did not delve into histone PTMs and their connection to histone degradation since we did not find studies directly relevant to age-related histone degradation. Outside the context of age-related histone degradation, we found studies that revealed the impact of histone PTMs that either reduced or enhanced degradation. This aspect has been reviewed briefly by Shmeuli et al [6]. A study in chronic obstructive pulmonary disease by Barrero et al., demonstrated that hyperacetylation renders H3.3 resistant to proteasomal degradation despite polyubiquitination of the histone [7]. Histone phosphorylation was also found to facilitate efficient polyubiquitination and proteasomal degradation of histones [8]. This degradation of histones resulted in transcriptional activation of tumor-promoting genes. Thus, PTMs can regulate histone degradation, but their role in aging remains unclear.

Comment 10: Overall, I find this review comprehensive and brilliantly written.

Response 10: We appreciate the positive feedback.

  1. Longo, V.D.; Kennedy, B.K. Sirtuins in aging and age-related disease. Cell 2006, 126, 257-268, doi:10.1016/j.cell.2006.07.002.
  2. Zhao, L.; Cao, J.; Hu, K.; He, X.; Yun, D.; Tong, T.; Han, L. Sirtuins and their Biological Relevance in Aging and Age-Related Diseases. Aging Dis 2020, 11, 927-945, doi:10.14336/AD.2019.0820.
  3. Kim, C.; Jin, J.; Ye, Z.; Jadhav, R.R.; Gustafson, C.E.; Hu, B.; Cao, W.; Tian, L.; Weyand, C.M.; Goronzy, J.J. Histone deficiency and accelerated replication stress in T cell aging. J Clin Invest 2021, 131, doi:10.1172/JCI143632.
  4. Liu, Y.; Wang, Y.; Yang, L.; Sun, F.; Li, S.; Wang, Y.; Zhang, G.A.; Dong, T.; Zhang, L.L.; Duan, W.; et al. The nucleolus functions as the compartment for histone H2B protein degradation. iScience 2021, 24, 102256, doi:10.1016/j.isci.2021.102256.
  5. Singh, R.K.; Kabbaj, M.H.; Paik, J.; Gunjan, A. Histone levels are regulated by phosphorylation and ubiquitylation-dependent proteolysis. Nat Cell Biol 2009, 11, 925-933, doi:10.1038/ncb1903.
  6. Shmueli, M.D.; Sheban, D.; Eisenberg-Lerner, A.; Merbl, Y. Histone degradation by the proteasome regulates chromatin and cellular plasticity. FEBS J 2022, 289, 3304-3316, doi:10.1111/febs.15903.
  7. Barrero, C.A.; Perez-Leal, O.; Aksoy, M.; Moncada, C.; Ji, R.; Lopez, Y.; Mallilankaraman, K.; Madesh, M.; Criner, G.J.; Kelsen, S.G.; et al. Histone 3.3 participates in a self-sustaining cascade of apoptosis that contributes to the progression of chronic obstructive pulmonary disease. Am J Respir Crit Care Med 2013, 188, 673-683, doi:10.1164/rccm.201302-0342OC.
  8. Xia, Y.; Yang, W.; Fa, M.; Li, X.; Wang, Y.; Jiang, Y.; Zheng, Y.; Lee, J.H.; Li, J.; Lu, Z. RNF8 mediates histone H3 ubiquitylation and promotes glycolysis and tumorigenesis. J Exp Med 2017, 214, 1843-1855, doi:10.1084/jem.20170015.

Round 2

Reviewer 2 Report

Comments and Suggestions for Authors

The authors provided the corrections and the quality of the manuscript has been improved. I recommend publishing. 

Author Response

Thank you for investing your time in reviewing our manuscript and for recommending it for publication.